**Resource**

# Functional annotation of noncoding mutations in cancer

Husen M Umer[1,2,*], Karolina Smolinska[1,*], Jan Komorowski[1,3,4,5], Claes Wadelius[6]

In a cancer genome, the noncoding sequence contains the vast majority of somatic mutations. While very few are expected to be cancer drivers, those affecting regulatory elements have the potential to have downstream effects on gene regulation that may contribute to cancer progression. To prioritize regulatory mutations, we screened somatic mutations in the Pan-Cancer Analysis of Whole Genomes cohort of 2,515 cancer genomes on individual bases to assess their potential regulatory roles in their respective cancer types. We found a highly significant enrichment of regulatory mutations associated with the deamination signature overlapping a CpG site in the CCAAT/Enhancer Binding Protein $\beta$ recognition sites in many cancer types. Overall, 5,749 mutated regulatory elements were identified in 1,844 tumor samples from 39 cohorts containing 11,962 candidate regulatory mutations. Our analysis indicated 20 or more regulatory mutations in 5.5% of the samples, and an overall average of six per tumor. Several recurrent elements were identified, and major cancer-related pathways were significantly enriched for genes nearby the mutated regulatory elements. Our results provide a detailed view of the role of regulatory elements in cancer genomes.

## Introduction

Whole genome sequencing (WGS) has been applied to investigate the molecular causes of cancer in several tumor types, but analysis has mainly focused on the protein-coding regions (Lawrence et al, 2013). However, because the majority of cancer genomic alterations are located in the noncoding regions, several recent studies have conducted detailed investigations on the impact of noncoding mutations in cancer (Hess et al, 2019; Liu et al, 2019; Kumar et al, 2020; Zhu et al, 2020). The impact of noncoding mutations has mainly been signified by recent discoveries of driver mutations in the *TERT* promoter, enhancers of *PAX5* and TAL and other regulatory elements (Huang et al, 2013; Fredriksson et al, 2014; Mansour et al, 2014; Weinhold et al, 2014; Katainen et al, 2015; Melton et al, 2015;

Nik-Zainal et al, 2016; Rheinbay et al, 2017). Recent efforts from the Pan-Cancer Analysis of Whole Genomes (PCAWG) project provided further evidence on the importance of noncoding mutations by identifying driver mutations in several noncoding regions including the 5′ region of *TP53* and 3′ untranslated regions of *NFKBIZ* and *TOB1* (Rheinbay et al, 2020). However, the large size of the noncoding genome in combination with sparse distribution of mutations in the genome has made it difficult to identify regulatory mutations in particular those that might not be highly recurrent (Sur & Taipale, 2016). To overcome this issue, two approaches are followed: (i) mutational burden tests (Lawrence et al, 2014; Lanzós et al, 2017; Zhu et al, 2020) that have limited power because they require large number of samples and (ii) functional tests (Fu et al, 2014; Lochovsky et al, 2015; Mularoni et al, 2016) investigating regulatory elements that are limited by the availability of biological experiments and tissue specificity of regulatory elements. As a result, the role of noncoding mutations has been underestimated since many of the previous statistical analyses have considered mutation recurrence or an insufficient set of functional elements (Weinhold et al, 2014; Rheinbay et al, 2017).

In addition, the complexity of the regulatory circuitry has hampered pinpointing functions of the noncoding mutations (Fu et al, 2014; Lochovsky et al, 2015; Mularoni et al, 2016). Also, cell type specificity of regulatory elements indicates that the same mutations may be functional only in certain cancer types depending on functionality of the regulatory elements. Recent large-scale efforts of the Encyclopedia of DNA Elements (ENCODE) and the Roadmap Epigenomics projects have provided functional annotations for identifying cell type–specific regulatory elements (ENCODE Project Consortium, 2012; Roadmap Epigenomics Consortium et al, 2015). Integrating these resources with cancer genomes provided in the PCAWG project (ICGC/TCGA Pan-Cancer Analysis Genomes Consortium, 2020) presents an opportunity to characterize the noncoding variants of individual tumor types.

We have used a unique approach to identify mutations that affect transcription factor (TF) motifs in tissues that are relevant to the tumor type. Our research design is motivated by the immediate impact of TFs on gene regulation. To this end, we used cell type–specific annotations to perform a comprehensive analysis of somatic

[1]Science for Life Laboratory, Department of Cell and Molecular Biology, Uppsala University, Uppsala, Sweden    [2]Department of Oncology-Pathology, Karolinska Institutet, Stockholm, Sweden    [3]Institute of Computer Science, Polish Academy of Sciences, Warsaw, Poland    [4]Swedish Collegium for Advanced Study, Uppsala, Sweden    [5]Washington National Primate Research Center, Seattle, WA, USA    [6]Science for Life Laboratory, Department of Immunology, Genetics and Pathology, Uppsala University, Uppsala, Sweden

Correspondence: claes.wadelius@igp.uu.se
*Husen M Umer and Karolina Smolinska contributed equally to this work

mutations in the PCAWG project and found many mutations in regulatory elements that are candidates to be functional.

# Results

## Functional annotations to prioritize noncoding mutations

We obtained high-quality somatic mutations from 2,515 tumor samples in 37 cancer types from the PCAWG project (See the Materials and Methods section and Fig S1). The vast majority of the mutations were located in the noncoding genome of which a fraction is expected to be functional. We reasoned that only mutations with regulatory functions in relevant cells and motifs should be considered. 85.8% of the mutations overlapped ENCODE DNaseI hypersensitive sites (DHSs) or TF binding sites (TFBSs) irrespective of cell type specificity (n = 105 cell lines) whereas only 3.7% of the mutations overlapped DHSs or TFBSs from cell lines matching the cancer types (Fig 1A). Nearly four million mutations overlapped TF motifs (Fig S2). However, only about 1.8% of them overlapped matching TF peaks and another 5.7% were marked by DHSs in matching cell lines (Fig 1B and D), whereas inactive chromatin states had a high load of somatic mutations (Fig 1C).

To quantify the importance of the chromatin signals, we applied a logistic regression model to assign a weight to each annotation (Fig S3). The model was trained on TF motifs within functional regulatory regions detected in massively parallel reporter assays (Ernst et al, 2016; Tewhey et al, 2016; Vockley et al, 2016; Umer et al, 2019 Preprint) (see the Materials and Methods section). Interestingly, matching TFBSs and DHSs from relevant cell lines had 90% and 79% positive predictive power in identifying functional motifs, respectively. Thus, a regulatory score was computed for each mutated motif using the funMotifs framework (Umer et al, 2019 Preprint). Also, we re-annotated the entire set of TF motifs using simulated mutations from 103 randomized sets to enable statistical evaluation of the regulatory scores.

We performed cohort-specific analysis to identify regulatory mutations in 44 cohorts (Rheinbay et al, 2020) (Fig 1D). The cohort set included a Pan-CCancer cohort combining all tumors except lymphoma and melanoma (ATELM). The statistical analyses to identify regulatory mutations and mutated regulatory elements were conducted on each cohort separately (Fig S3). Because multiple neighboring mutated motifs may affect the same regulatory element, we defined mutated elements by merging mutated motifs within 200 bp (see the Materials and Methods section). To account for local hypermutated regions, we assessed the significance of the regulatory score of each element by comparing it to a local background distribution of simulated elements within a 50 kb window and only those with empirical P-value < 0.05 were retained (Table S1 and Fig S4). However, because of the lack of a well-designed simulated mutation dataset that recapitulates mutation occurrence in various cancer types, we enforced additional conditions on the significant elements. To this end, the mutation significance level of each element was evaluated using Active-DriverWGS and those with false discovery rate (FDR) < 0.05 were kept (see the Materials and Methods section) (Zhu et al, 2020). To ensure

the regulatory role of the mutated elements, we also conditioned the elements to contain at least one regulatory mutation. As depicted in Fig 1E, mutations in the significant elements with high impact on TF motifs and overlapping chromatin signals in a matching tissue type were defined as regulatory mutations (see the Materials and Methods section). Moreover, to avoid removing mutations due to the lack of motif predictions for TFs or the lack of ChIP-seq experiments in the designated cell lines, we extended the mutated elements to be of a typical regulatory element size (200 bp) and additional mutations that were located in the regulatory elements were retained. The element size was based on the observation that most TFs in the same regulatory element bind within 200 bp (Diamanti et al, 2016). The final set of recurrent regulatory elements was conditioned to be mutated in at least three samples. These analyses provided the set of significant mutated elements that have potential regulatory roles per cohort. Across the 44 cohorts 98,302 regulatory mutations were identified (Table S2). Initially, FOXP1, IRF1, and ZNF263 each had more than one million predicted motifs and therefore they were most enriched for mutations in comparison to other TFs. However, after conditioning on the functional annotations defined here only a small fraction of the mutated motifs remained (Fig S5).

## Mutational signatures in TFBSs

To characterize the mutational processes underlying mutations in TFBSs (Alexandrov et al, 2013, 2020), we examined differences in the mutational context of regulatory mutations identified above and the remaining mutations per cancer type (see the Materials and Methods section). Overall, 46 single-base substitution (SBS) and double-base substitution (DBS) signatures had a significantly different contribution in at least one cancer type (Kolmogorov–Smirnov two-sided test, P-value < 0.05, Fig 2). Interestingly, enrichment of SBS1, SBS30, and SBS39 was higher for regulatory mutations, whereas enrichment of SBS8, SB12, and SBS36 was lower in at least 15 cancer types. These signatures have previously been reported as the most common signatures across the tumors in PCAWG (Alexandrov et al, 2020). Notably, APOBEC associated signatures (SBS2, SBS13, and SBS69) were significantly less enriched for regulatory mutations in many cancer types (Rheinbay et al, 2020). We also measured the cosine similarity between trinucleotide context profiles of the regulatory mutations and COSMIC SBS signatures v3 (see the Materials and Methods section) (Tate et al, 2019). The cosine similarity levels were low for most of the SBS signatures (mean cosine similarity = 0.26, standard deviation = 0.17). The differences in the signatures associated with regulatory mutations and other mutations may suggest that different processes are involved in creating mutations in TFBSs (Fig 2).

Moreover, in skin melanomas, regulatory mutations had a significantly lower contribution from UV-light associated signatures: SBS7a, SBS7c, SBS38, SBS65, SBS67, SBS75, and DBS13 (Rheinbay et al, 2020). Furthermore, the cosine similarity between the context of regulatory mutations in skin melanomas and the COSMIC signatures was below 0.42 for all SBS signatures (median value = 0.06, Fig 2) except SBS7a and SBS7b that are associated to UV-light (Fig S6, see the Materials and Methods section).

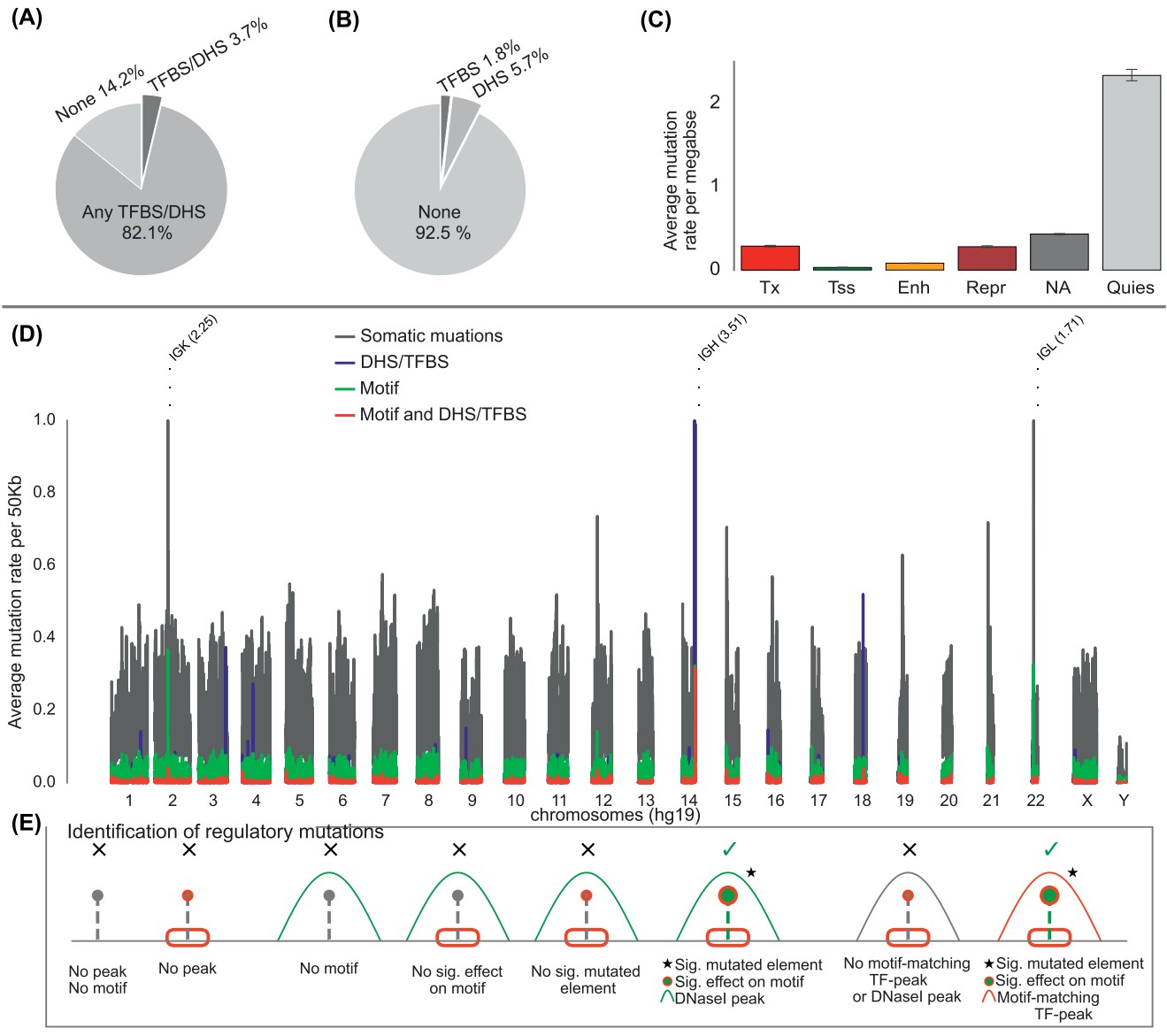

**Figure 1. Mutation rates and selection of regulatory candidates.**
**(A)** Somatic mutations overlapping transcription factor binding sites (TFBSs) or DNaseI hypersensitive sites (DHSs) from cell lines matching the corresponding cancer types (3.7%) and other cell lines (82.1%). "None" denotes mutations that do not overlap any TFBS or DHS. **(B)** Overlap of somatic mutations in TF motifs with TFBSs or DHSs from cell lines matching the corresponding cancer types. "None" denotes mutations that do not overlap any TFBS or DHS from the respective cell lines. **(C)** Enrichment of mutations per chromatin state in 1 Mb windows across the genome. The average per window is taken as the total number of mutations divided by the total number of samples (2,515). The 15-model chromatin states are collapsed by representing Tx, TxFlnk, TxWk as Tx; TssA, TssAFlnk, TssBiv, and BivFlnk as Tss; Enh, EnhG, and EnhBiv as Enh; ReprPC and ReprPCWk as Repr; Quies, ZNF/Rpts, and Het as Quies. **(D)** Rate of somatic mutations per chromosome. The rate is calculated as the total number of mutations in 50 kb windows divided by the total number of samples (2,515). The colors exhibit mutations that (i) overlap TF motifs and matching TFBSs or DHSs from respective cell lines (red), (ii) overlap motifs but no DHS/TFBS is found from the respective cell line (green), (iii) overlap DHSs/TFBSs from respective cell lines but have no overlap with TF motifs (blue), (iv) have no overlap with TF motifs or TFBSs/DHSs from the respective cell lines (gray). The labeled bars represent the hyper-mutated loci that on average contain more than one mutation per 50 kb. **(E)** Strategy applied to detect regulatory mutations. The red box represents TF motifs, the red peak represents binding site of a TF matching the motif, the black peak represents binding site of a TF not matching the motif, and the green peaks represent DHSs. The star indicates the mutation located in a significant regulatory element.

## Mutational patterns in TFBSs

Next, we asked whether regulatory mutations target motifs of any specific TF (Fig 3A). In agreement with previous studies, we found CTCF as the most mutated TF in most of the cohorts. Its motifs were mostly mutated in digestive tract tumors (3,784 CTCF mutated motifs, P-value = 0.0097), esophagus cancers (1,154, P-value = 0.0097), liver-HCC (1,570, P-value = 0.0097), and skin melanomas (1,016, P-value = 0.0097, Fig 3B) (Katainen et al, 2015; Sabarinathan et al, 2016; Umer et al, 2016). In the ATELM cohort, 5,968 CTCF motifs were mutated (P-value = 0.0097); 25.5% and 18.3% of those were in liver-HCC and in esophagus cancers, respectively. Interestingly, the mutational profile

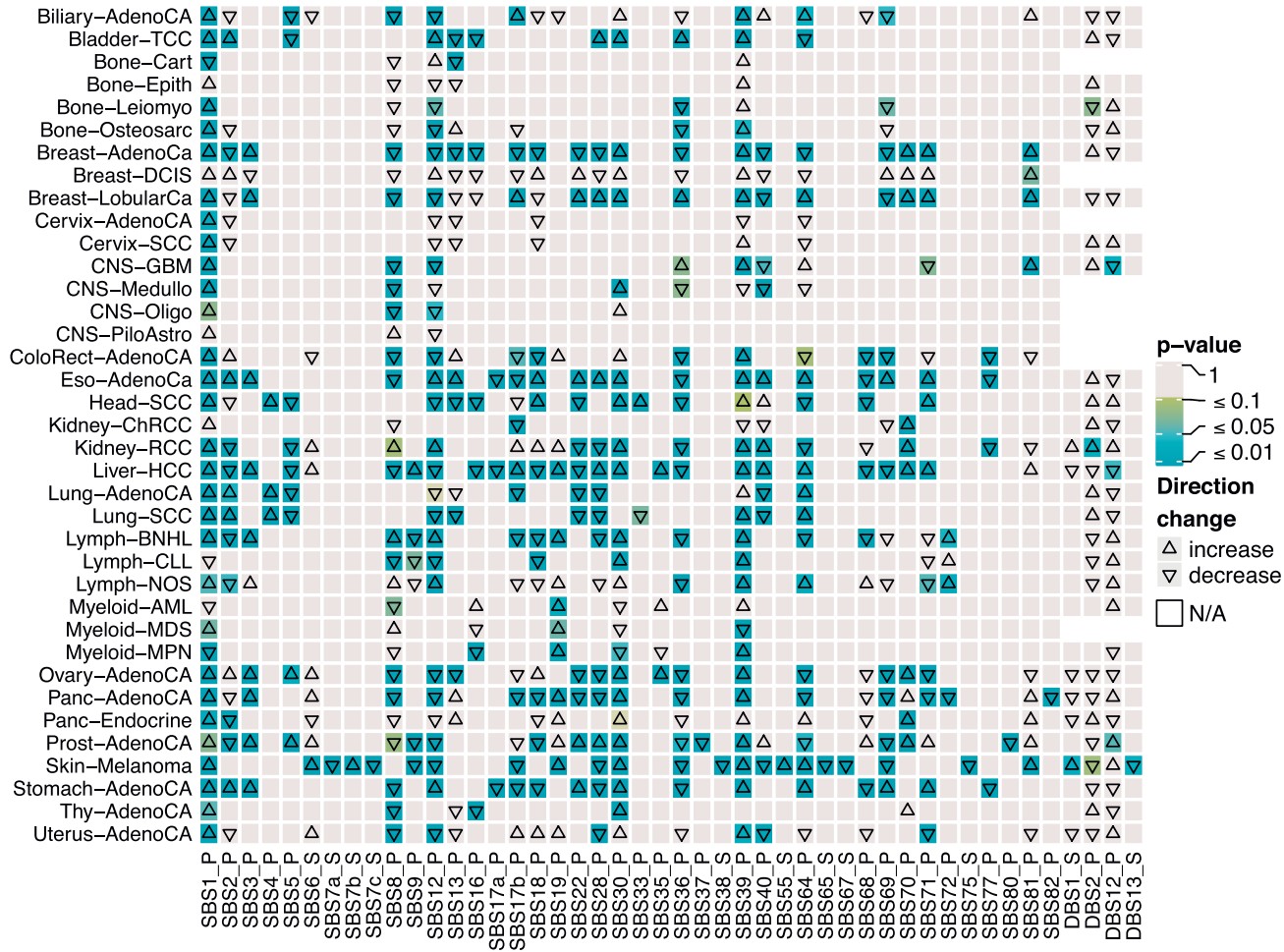

**Figure 2. Comparison of mutational signature distribution between regulatory mutations and all mutations across 37 cancer types.**
The orientation of triangles shows a direction of the signature contribution change from mutations to regulatory mutations in the same cancer type. The color of squares indicates the significance of contribution change evaluated using a Kolmogorov–Smirnov two-sided test. The plot presents a fraction of significantly different signatures (P-value < 0.05) in at least one cancer type.

of regulatory mutations in CTCF motif showed an overrepresentation of SBS17b signature in esophagus, lymph-NOS, colorectal, and stomach cancers (Fig S7). Also, the cosine similarity between the profile context of the CTCF regulatory mutations in these cancer types and SBS17b was between 0.6 and 0.72. The etiology of the SBS17b signature is unknown, nevertheless the association of the oxidative damage of DNA and SBS17b in gastrointestinal cancers has been reported previously (Kauppi et al, 2016; Tomkova et al, 2018). Sabarinathan et al (2016) have attributed mutations at active motifs of CTCF and other TFs in melanomas to impairment of the DNA repair machinery. However, even though CTCF bindings in GM12878 are almost as abundant as in HepG2 (human liver carcinoma cells) and keratinocytes, the mutation rate in CTCF motifs was much lower in the lymphomas compared to liver cancers and melanomas. Notably, signatures SBS7a and SBS7b that are characteristic for the UV-light exposure were enriched for CTCF mutations in melanomas (Fig S7) and had the highest cosine similarity (SBS7a: 0.50 and SBS7b: 0.82) among the other UV-light signatures. Finally, in lung cancers (Lung-AdenoCA and Lung-SCC) SBS4 signature, which is associated to

tobacco smoking, had the highest contribution (Letouzé et al, 2017). Overall, different spectra of mutational signatures were observed at CTCF motifs in different cancer types.

Interestingly, position 9 of the CTCF motif was frequently mutated (Fig 3F), particularly in liver, esophagus and stomach, which is in agreement with our previous findings (Umer et al, 2016). The mutational profile of regulatory mutations identified at position 9 clearly indicated the elevation of C[T>G]N mutations, which is characteristic for the SBS17b signature (cosine similarity = 0.47, Fig S8). In contrast, positions 13 and 14 of the CTCF motif were significantly mutated in melanomas, indicating different mutational signatures associations at CTCF motifs in gastrointestinal cancers and melanomas.

Furthermore, motifs of CCAAT/Enhancer Binding Protein $\beta$ (CEBPB) were found to have a significant number of mutations in many cohorts (Fig 3B). In the ATELM cohort, 1,436 CEBPB motifs were mutated (P-value = 0.0097; fold-enrichment = 1.7 over the background enrichment), which was higher than the other significantly mutated TF motifs, that is, CEBPBG (654 motifs), CEBPD (336), ZBTB33

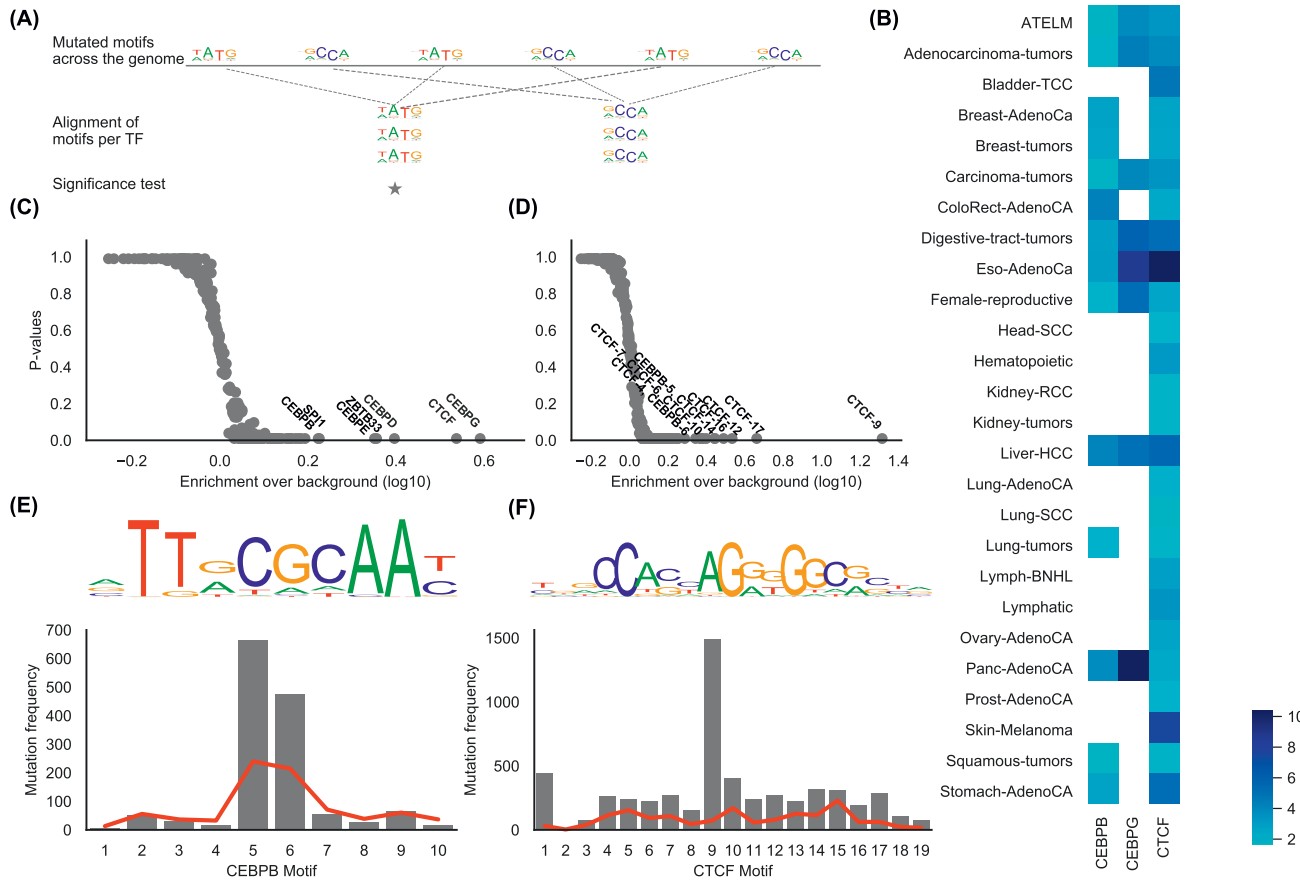

**Figure 3. CCAAT/enhancer binding protein β and CTCF are the most affected TFs genome-wide.**
**(A)** Strategy applied to align mutated motifs for computing mutational load per TF. The star indicates a significant score over the background mutational load. **(B)** Heat map showing enrichment of mutations in motifs per TF over the background enrichment estimated from 103 simulated sets. The color range is based on robust quantiles centered on 1. **(C)** Cumulative enrichment of mutations in motifs per TF. Labels indicate TFs with enrichment over the background higher than 1.6 and *P*-value < 0.01. Labels were merged based on the decreasing enrichment value. **(D)** Cumulative enrichment of mutations per motif position per TF. Labels indicate TFs with enrichment over the background higher than 2 and *P*-value < 0.01. **(E, F)** Enrichment of mutations in CCAAT/enhancer binding protein β motif and (F) in CTCF motif. The upper panel shows the motif logo, and the red line shows enrichment that was observed in the simulated sets.

(409), CEBPE (323), and SPI1 (866), See Fig 3C. Furthermore, 17.1% of the motifs were mutated in two or more samples. CEBPB has been shown to regulate survival, apoptosis and senescence (Zhu et al, 2002; Sterneck et al, 2006; Kurzejamska et al, 2014; Barakat et al, 2015; Tamura et al, 2015) and increased mutation rate in its motif has been reported (Melton et al, 2015; Vorontsov et al, 2016). Elevation in expression of CEBPB has been linked with progression of glioblastoma, lymphoma, and breast cancer (van de Vijver et al, 2002; Jundt et al, 2005; Homma et al, 2006; Piva et al, 2006).

Notably, positions five and six that define a 5′—C—phosphate—G—3′ (CpG) site within the CEBPB motif were most significantly mutated motif positions of all TFs excluding CTCF in the ATELM cohort (Fig 3D). As expected, the majority of the mutations at position five and six of the motif were C>T (94.2%) and G>A (88.4%), respectively (Fig 3E). C>T mutations are characteristic of the SBS1 signature. Thus, the mutational profile of CEBPB motifs at position five and six could clearly be explained by the SBS1 signature with a cosine similarity of 0.8 compared to 0.52 for the remaining mutations (Figs S9 and S10).

Because SBS1 is characterized by deamination of 5-methylcytosine to thymine in double stranded DNA that mainly causes C to

T transitions in CpG contexts, we investigated the methylation pattern within CEBPB motifs. Analysis of methylation data from the same tumors indicated a significantly lower mutation rate at methylated cytosine-guanine (CG) dinucleotides within CEBPB motifs in comparison to neighboring methylated CG dinucleotides (Fisher's exact test, *P*-value = 0.012, odds ratio = 3.83, see the Materials and Methods section). Finally, we observed that the active motifs of CEBPB are significantly less methylated genome-wide than inactive motifs and still there is an abundance of mutations at position 5 (Mann–Whitney test, *P*-value < 0.01, see the Materials and Methods section). Sayeed et al (2015) have performed EMSA experiments showing subtle effects of methylation in the CG site on CEBPB binding. Notably, they found that 5-methylcytosine at the CG site enhances binding of CEBPB. Enhanced binding of CEBPB with C>T mutations at base 5 of the motif was recently confirmed by Ershova et al (2020) *Preprint*.

Besides SBS1, the most common mutational signature in regulatory mutations at CEBPB motifs across most cancer types was SBS5, which is characterized by enrichment of T>C and C>T mutations. The etiology of SBS5 is unknown; however, SBS1 and SBS5

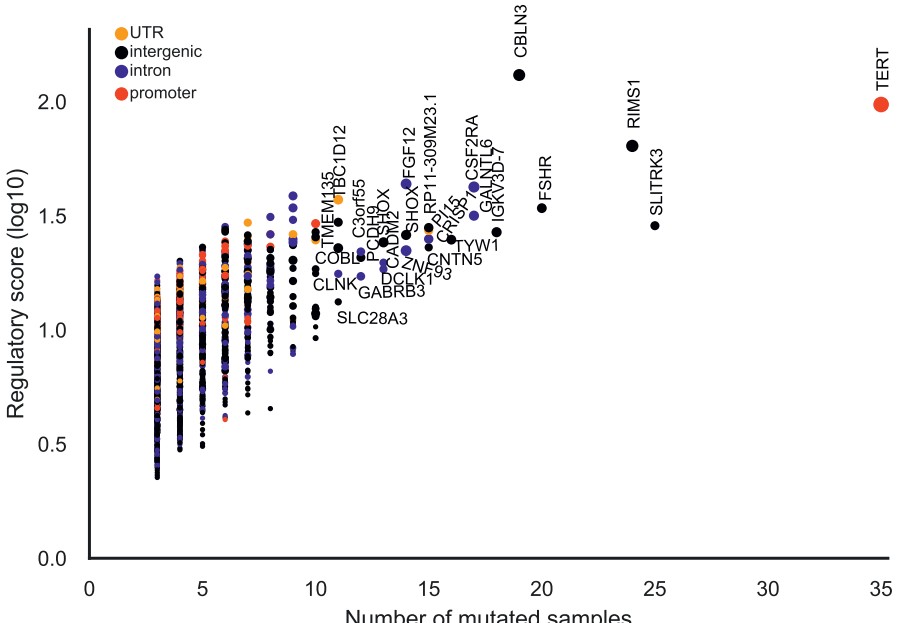

**Figure 4. Significantly functional mutated regulatory elements.**
The dots denote mutated elements detected in all cohorts excluding the melanoma, lymphoma cohorts. The colors represent feature types. The marker sizes represent the number of samples that have regulatory mutations in the element. Gene labels are given for elements that are mutated in more than 10 samples.

signatures are highly age-correlated (Alexandrov et al, 2020). The cosine similarity between the mutational profile of CEBPB motifs and the SBS5 signature was noticeably higher for the mutations outside positions 5 and 6 (cosine similarity = 0.64) than mutations at positions 5 and 6 of the CEBPB motif (cosine similarity = 0.11, Figs S9 and S10). The lower enrichment of SBSB5 signature and lower methylation level at position 5 and 6 of the CEBPB motif may indicate other mechanisms than clock-wise mutational processes. Finally, by analyzing RNA-seq from matching samples, we found that expression of genes in the proximity of CEPB motifs (2 kb) was significantly dysregulated in the mutated samples (Wilcoxon signed-rank test, *P*-value < 0.05, see the Materials and Methods section).

## Mutated regulatory elements in cancer

We investigated the significant regulatory elements to identify recurrent events in cancer. To this end, we merged overlapping significant elements identified across the cohorts excluding lymphoma and melanoma cohorts. The elements that were mutated in at least three samples were retained to obtain the final list of recurrent elements. Across the 39 cohorts, 5,749 recurrent regulatory elements were detected (Table S3). The elements contained 178,978 mutations of which 11,962 were regulatory mutations that overlapped 28,292 potentially functional motifs. Our results showed an average of six regulatory mutations per tumor sample. Also, 5.5% percent of the samples had 20 or more regulatory mutations whereas 66.2% had six or fewer (Fig S11). The majority of the elements were either intergenic or intronic; 60.7% and 27.6%, respectively, whereas 6.9% were in promoters, and the remaining 4.8% were located in UTRs.

Overall, 16.7% and 10.6% of the regulatory mutations were enriched in transcription start site (TSS) and enhancer chromatin states, respectively. Notably, 62.1% of the tumor samples had regulatory mutations in TSS regions.

At TSSs, motifs of SP2, EGR1, and SP1 were most frequently mutated; 5.7%, 12.9% and 11.9% of the tumors, respectively (Fig S12). EGR1 is a direct regulator of many tumor suppressor genes (Baron et al, 2006), also SP TF family members (SP1 and SP2) are involved in gene regulation in tumors (Archer, 2011). SP1 is a TF that can activate, and repress transcription, and is involved in many cellular processes including cell growth, apoptosis, and response to DNA damage. In contrast, quiescent regions had the highest regulatory mutation enrichment in CTCF, ZNF263, CEBPD, and CEBPB motifs; 31.6%, 22.1%, 20.9%, and 20.6% of the tumors, respectively (Fig S12).

As it has previously been reported, the well-known *TERT* promoter was the most mutated element (Fig 4) (Fredriksson et al, 2014; Rheinbay et al, 2020). It also had the highest functionality score (*P*-value = 1.3 × 10⁻¹⁶) and was mutated in 35 tumor samples. C>T mutations overlapped position 11 in 16 EGR1 motifs. EGR1 has been shown to downregulate TERT expression and it has been suggested as a tumor suppressor (Mittelbronn et al, 2009). Creation of a de novo ETS motif at the EGR1 motif locus has been shown to up-regulate TERT expression (Pagel et al, 2012). Although the *TERT* promoter is reported as the most recurrent element in our analysis, creation of the de novo ETS motif is not reported because our analysis is based on pre-annotated motifs. Therefore, additional candidate mutations creating de novo motifs will identify further candidate elements.

Moreover, among the most recurrent elements, there were five intergenic elements mutated in more than 18 samples (Table S3). Recurrent elements were also found in introns of *CSF2RA* (*n* = 17 samples), *GALNTL6* (17), *CRISP1* (15), *FGF12* (14), *ZNF93* (14), DCLK1 (13), *CADM2* (13), *GABRB3* (12), *C3orf55* (12), *IL1RAPL1* (12), and *CLNK* (11) (Fig 4). *CSF2RA* and *FGF12* are part of the Kyoto Encyclopedia of Genes and Genomes (KEGG) cancer pathway. *CSF2RA* encodes for CD116 which is a key cytokine associated to proliferation, survival, and differentiation of myeloid cells and its deficiency has been shown to enhance t(8;21) leukemia (Hansen et al, 2008; Matsuura et al, 2012).

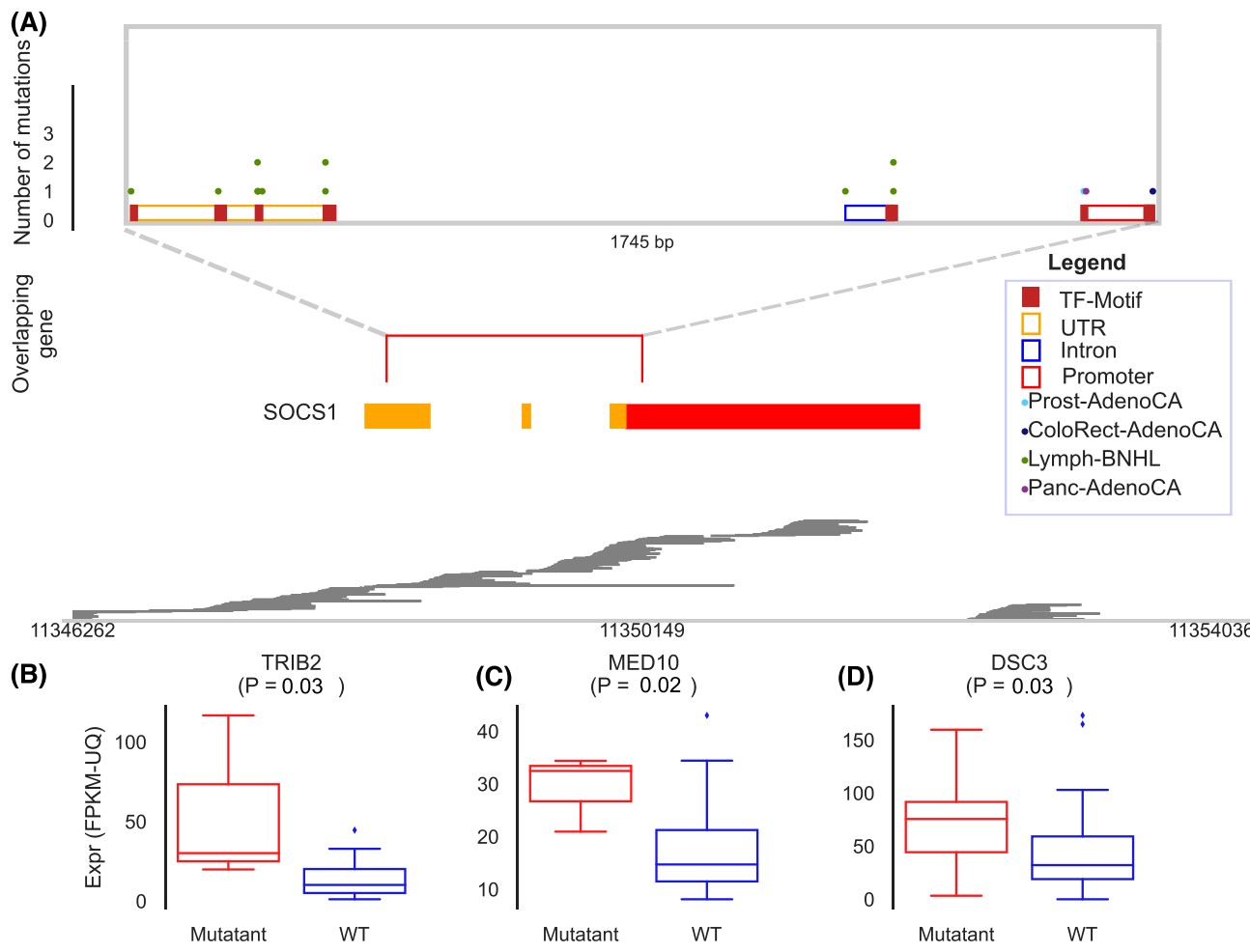

**Figure 5. Differential expression of genes associated to highly recurrent mutated elements.**
**(A)** The three highly mutated elements are found in the 5′UTR, intron and proximal-promoter of the *SOCS1* oncogene. Only mutations found in potential regulatory elements are shown. The dots in the upper panel represent mutations. The boxes in the middle panel represent gene exons. TF peaks from ChIP-seq data in GM12878 are shown in the lower panel. **(B, C, D)** Box plots of expression levels (FPKM and upper quantile normalized) for (B) TRIB2 in bladder samples, (C) MED10 in bladder samples and (D) DSC3 in lung samples with and without mutations in their associated elements. The central line is the median, the box boundaries are the 25th and 75th quartiles and the individual samples are indicated with dots. The *P*-values were calculated using a permutation-based *t* test approach excluding samples that had significant copy number variants at the loci (see the Materials and Methods section).

Previous analyses of regulatory elements have reported the 5′ UTR of *WDR74* as a significantly mutated element (Weinhold et al, 2014). However, in our analysis despite a high number of somatic mutations that resided in the element (*n* = 31 mutations), only two of them were identified as regulatory mutations (*P*-value = 0.01). Also, the mutations in the element were scattered across motifs of multiple TFs. Therefore, most of the mutations in this element have no effect on regulation of *WDR74*. Notably, Weinhold et al (2014) did not find altered transcription levels of WDR74 in mutated tumor samples, and Rheinbay et al (2020) filtered out the elements associated to *WDR74* owing to mapping issues. These results show that functional annotations are needed to overcome the limits of mutational burden tests because not all mutations in a regulatory element are functional. This also indicates that our method is complementary to previous methods for finding coding and noncoding candidates.

We also found multiple genes with several recurrent mutated elements in various gene regions, mostly in introns. For instance,

Suppressor Of Cytokine Signalling 1 (*SOCS1*) had a recurrent element in its proximal promoter in prostate, colorectal and pancreatic cancer samples (FDR < 0.00225, Fig 5A). It also had recurrent elements in 5′ UTR and the first intron in 9 lymphoma samples (Fig S13). *SOCS1* is a known oncogene in many cancer types including lymphoma (Beaurivage et al, 2016; Chevrier et al, 2017; Khan et al, 2020; Liu et al, 2003). Interestingly, the intergenic region nearby *TRIB2* was highly enriched for mutated elements (three elements with FDR < 0.0014), and its expression was significantly increased in the mutant bladder cancer samples compared to the samples that lacked mutations in the associated elements excluding samples that had copy number variants (CNVs) at the loci (Fig 5B, see the Materials and Methods section). Expression of TRIB2 is increased in melanoma, colon and pancreatic cancer which leads to impaired therapeutic response and poor clinical outcome (Hill et al, 2017). Furthermore, *MED10* had multiple hotspots in its promoter, and an associated intergenic region (FDR < 0.022). The elements were

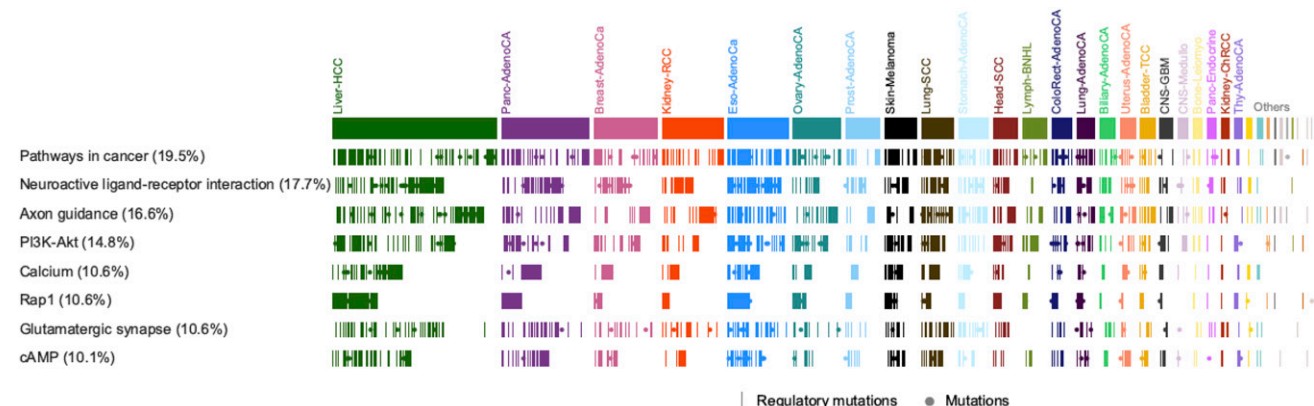

**Figure 6. Most enriched pathways.**
The percentage of samples that are mutated per pathway are shown next to the pathway labels. The colors represent cancer types of the mutated samples. Cancer types that had <10 samples mutated are shown as *Others*. Samples that had regulatory mutations in a gene associated to the respective pathway are indicated with a bar line and those that had a mutation in the identified mutated elements but lacked direct-functionality evidence are shown as dots.

mostly found in kidney, pancreas, lung, head, esophagus and bladder cancers. *MED10* was significantly up-regulated in bladder mutant samples (*P*-value = 0.02, Fig 5C). Increased expression of MED10 is associated with poor outcome in liver, renal cancer and glioma (*MED10* prognostics from https://www.proteinatlas.org/ENSG00000133398-MED10, Uhlén et al, 2015). Finally, *DSC3* also had multiple recurrent elements and it was up-regulated in the mutant samples from lung cancer (*P*-value = 0.01, Fig 5D).

To avoid bias from hyper-mutated elements and samples, we investigated the lymphoma and melanoma cohorts separately from the other 39 cohorts. In the lymphatic system cohorts, only 23 regulatory elements were identified and five of the elements overlapped mutated elements from the non-lymphatic system cohorts (Table S4 and Fig S13). The Immunoglobulin Heavy (IGH) locus was the most mutated element (Fig S13). There were three highly recurrent mutated elements overlapping multiple IGH segments including *IGHJ6*, *IGHG3*, *IGHG1*, and *IGHM* that were mutated in 97 Lymph-BNHLs, 44 Lymph-CLLs, and 2 Lymph-NOS. Interestingly, Bach1::Mafk and TCF12 motif were most enriched for mutations at this hotspot and were mutated in 46 and 44 samples, respectively. However, because of the high mutation rates in the immunoglobin loci, special care needs to be taken for further consideration of these elements. Also, in the melanoma cohort, we identified 151 mutated elements, and 36.6% of them overlapped mutated elements from the other cohorts (Table S5 and Fig S14).

### Enrichment of cancer genes and pathways in mutated regulatory elements

We sought to investigate the effects of the recurrent elements identified across the 39 cohorts in which the lymphoma and melanoma samples were excluded (Table S3). Overall, we identified 362 genes that were enriched for mutations in 10 or more samples by combining genes within 2 kb upstream/downstream of the mutated elements. The closest gene was considered when genes were not present in the 2 kb proximity from the mutated elements (see the Materials and Methods section, Table S6). The list

comprised genes with established roles in cancer such as *CTNNA2* (n = 54 samples), *TERT* (38), *CTNNA3* (37), *LRP1B* (33), *PTPRT* (33), *CDH11* (28), *FGF12* (24), *DCC* (21), *FGF13* (21), and *RUNX1T1* (21). Our analysis also indicated other genes that had high enrichment for mutations (more than 65 samples) in their mutated elements including *BRINP3*, *CNTNAP2*, *CDH9*, *CSMD3*, *BAI3*, *KLHL1*, *NOVA1*, *LINC000273*, *GALNTL6*, and *SLITRK5*. We further investigated overrepresentation of pathways for the genes in which their associated regulatory elements were mutated in at least three samples. Interestingly, the major KEGG Cancer Pathways (KEGG ID: hsa05200) was significantly enriched and it was most mutated (FDR = $6.0 \times 10^{-5}$, n = 462 tumors, hypergeometric test). PI3K-Akt, Rap1, and cAMP signaling pathways were among the top mutated ones with significant FDRs < 0.0015 (Table S7). The most enriched pathways (FDR < 0.05) that were mutated in more than 10% of the samples are shown in Fig 6.

## Discussion

We focused here on the potential functional mutations for evaluating mutational burden in TF motifs and for identifying candidate functional elements in cancer. Overall, we processed more than 25 million observed mutations, 2.6 billion simulated mutations and millions of functional annotation tracks. Our study is the largest analysis of noncoding mutations in TF motifs to date. A rigorous annotation pipeline was applied to identify functional TF motifs in cell types matching the corresponding cancer. The cancer type–specific annotations led to the identification of regulatory mutations that had a significant impact on TF motifs. Mutational signature analysis indicated different mutational processes underlying regulatory mutations in comparison to other mutations. Interestingly, APOBEC-associated signatures that are enriched across cancer types were significantly lower enriched for regulatory mutations in a combined cancer type cohort. Also, regulatory mutations in melanoma samples had significantly lower contributions to many of the UV-light signatures that are usually observed in melanoma.

Two approaches were applied to identify recurrent events that may point to positive selection in cancer. The first approach was to identify recurrent mutation patterns by accounting for the sparse distribution of noncoding mutations. In addition to CTCF, we identified mutations in motifs of CEBPB across many cancer types and, in particular, mutations affecting the CpG site of the motif that were attributable to the SBS1 signature. It has been shown that methylation improves binding of the CEBP family so mutating the CpG may affect binding by both changing the motif and preventing DNA methylation (Vinson & Chatterjee, 2012; Sayeed et al, 2015; Yin et al, 2017). CEBP proteins have also been found to be critical for activation of tissue specific promoters in many tissues (Buck et al, 2001; Rishi et al, 2010). Therefore, mutations at these sites are highly suggestive of potential effects on the target genes. Most of the enriched cancer types had a similar mutation rates in CEBPB motifs whereas CTCF motifs were highly mutated particularly in gastrointestinal cancers and melanomas, whereas lymphoma cohorts were not enriched for recurrent patterns.

Our second approach was to identify functional regulatory elements that were recurrently mutated. Performing meta-cohort analysis by integrating multiple cancer types was key to identify highly mutated elements across the cancer types. We observed a significant enrichment of cancer-associated genes near the identified recurrent elements indicating that cancer genes lacking coding mutations may be differentially expressed because of mutations in their regulatory elements. Furthermore, many cancer pathways were significantly enriched for genes that were associated to the mutated regulatory elements. These findings suggest a role of noncoding mutations in cancer, especially when looking for functional mutations and not only drivers.

We expect a higher number of candidate functional mutations to be eventually found since we have only included established TF motifs in our analysis. Therefore, future analysis investigating de novo motifs will identify further candidate mutations that enhance binding of TFs. Also, discovery of further TF models and generating new datasets for cell types or TFs that are not yet assayed will improve identification of regulatory elements. Finally, the power to detect regulatory mutations is smaller than for finding coding mutations. The reason is that coding sequences mainly are the same between cancer types whereas enhancer and other regulatory elements differ between cell types limiting the possibility to find recurrent events. Therefore, larger studies are needed to find all functional and recurrent elements that are mutated in cancer.

# Materials and Methods

### Somatic mutation collection

Single nucleotide variants (SNVs) and short insertions and deletions <100 bp were retrieved from the catalog of somatic mutations of 2,577 white-listed samples in the PCAWG project. Somatic mutations that were concordantly detected by at least two of the four mutation calling pipelines were kept. Samples with more than 100,000 somatic mutations were marked as hypermutated, thus removed from the collection ($n$ = 63). Multiple nearby SNVs from the

same samples were collapsed into dinucleotide (DNP), trinucleotide (TNP), and oligonucleotide (ONP) polymorphisms.

### Mutation simulation

Simulated sets ($n$ = 100) were generated following the Sanger simulation strategy (Rheinbay et al, 2020). Briefly, each mutation was shuffled to a random position with an identical trinucleotide context. To get a mutation rate similar to the observed dataset, the random position was selected within a 50 kb window and 50 bp away from the original mutation position. In addition, we also included the three simulation sets generated in the driver paper (Rheinbay et al, 2020).

### Motif identification and annotation

We used the funMotifs framework to annotate TF motifs (Umer et al, 2019 Preprint). Briefly, TFBSs and DHSs in 105 cell lines and tissues were gathered from www.encodeproject.org (accessed on 29 November 2016) (ENCODE Project Consortium, 2012). Position frequency matrices were downloaded for 519 TFs from JASPAR 2016 (Mathelier et al, 2016). The ENCODE regions were merged and scanned for significant $k$-mers ($P$-value < 1 × 10$^{-4}$) using FIMO (Grant et al, 2011). Only top significant motif instances ($n$ = 85,459,976) covering 14% of the genome were considered within each TF motif set ($Z$-score > 1). The $Z$-score was based on the scores of all identified motif instances of each TF, that is, instances with a score one standard deviation above the mean (Umer et al, 2019 Preprint).

Next, genomic datasets were collected from various assays and cell lines. For each tumor type, we gathered annotations from the cell lines most similar to the respective tumor type (Table S8). In cases where no data were available for any closely related cell line, we imputed signals from the other available cell lines only when the annotations were available in at least four cell lines. For categorical annotations, the most common label was assigned to fill the missing value, whereas for numeric features the arithmetic mean was taken from the tissues that had the annotations. The annotations were TF binding assays and DHSs from ENCODE (ENCODE Project Consortium, 2012), replication domains from Liu et al (2016), TF expression from GTEx V6p (Carithers & Moore, 2015), CAGE peaks from FANTOM (Andersson et al, 2014; FANTOM Consortium and the RIKEN PMI and CLST (DGT), 2014), and chromatin states from RoadMap Epigenomics (Roadmap Epigenomics Consortium et al, 2015) (Table S9).

### Mutation scoring mechanism

TF motifs were annotated for each tumor type based on annotations of cells most similar to the respective tumor type. Weights for the annotation features were obtained from a logistic regression model that was trained on functional elements from massively parallel reporter assays in the funMotifs framework (Umer et al, 2019 Preprint). Only features with a positive log odds ratio were used. TF motifs overlapping somatic mutations were scored based on weights of the overlapping annotations in the corresponding tumor type, and changes in the motif matching score (Umer et al, 2019 Preprint). The change in motif matching score was computed

as the absolute value of the reference allele minus the mutant allele in the corresponding Position frequency matrix (Fu et al, 2014).

### Cohort specific analysis

To account for variable mutation rates across the cancer types, the observed and the simulated somatic mutations were distributed onto 44 cohorts (Table S10). 29 of the cohorts were tumor type specific and the remaining ones included combinations of tumor types. We discarded all mutated motifs of TFs not expressed in cell lines respective to the cancer types. Next, regulatory elements and mutations were identified from each cohort separately. The cohort specific analysis allowed us to account for the high mutation rates in melanomas as well as high mutation rates at the immunoglobulin loci in lymphomas. Melanomas showed the highest variation across the tumor samples whereas lymphatic system tumors had large number of mutations in the immunoglobulin loci.

### Identification of mutated elements

In each cohort, mutated motifs within 200 bp were merged to form mutated genomic elements and the mean of the motif scores was assigned to the corresponding elements. To account for hypermutated regions, the score of each element was compared to a neighborhood background distribution of scores. For each element an empirical *P*-value was calculated by comparing the regulatory score of the mutated element to the scores of elements in the simulated sets within a 50 kb window. The 50-kb window was used because the simulated sets were generated by shuffling the observed mutations within the same distance. Finally, elements with a *P*-value < 0.05 were selected to have significant regulatory scores.

We compared the distribution of the *P*-values to the theoretical distribution to ensure the statistics are not inflated. The theoretical distribution was obtained based on randomly selected simulated elements from the same window as the observed element. The score of the selected element was compared to the other simulated elements in the window to compute a *P*-value. The resulting *P*-values across the windows were collected to generate the final background distribution and compute the quantile–quantile plots shown in Fig S4.

In addition, we applied ActiveDriverWGS to evaluate the mutation significance level of the elements that had significant regulatory score from the previous step (Zhu et al, 2020). The mutation enrichment was estimated based on all annotated somatic mutations. The background model was created using Poisson generalized linear regression based on the mutations presented in the ±50 kb window around the elements. For SNVs, the trinucleotide composition was treated as cofactors. Importantly, ActiveDriverWGS excludes potentially hypermutated samples with more than 30 mutations per megabase. The significantly mutated elements (FDR < 0.05) were retained except those that were in coding regions.

Following this strategy, mutated elements were obtained for all the cohorts. The tumor meta-cohorts that contained samples from all tumor types except lymphoma and melanoma samples enabled discovery of mutated elements across all cancer types.

Mutations within the significantly mutated elements were defined as regulatory mutations if they changed the matching score of

a functional TF motif within the element by more than 0.3 contrasting the background nucleotide frequency on a single position which is 0.25 per nucleotide. Also, the regulatory mutations had to overlap either a matching TF peak or DNase1 from a tissue corresponding to the same tumor type.

Thus, only significant elements that contained regulatory mutations were retained and those without any regulatory mutation were discarded even when they were mutated in many samples. Finally, we extended the significant elements to 200 bp and intersected them with the entire set of mutations to retain any additional mutation that was removed because of the lack of overlapping TF motifs.

### Mutational signature analysis

Mutational signatures were provided by the PCAWG-7 group (Synapse: syn11738306: SignatureAnalyzer_COMPOSITE.SBS_signature_probability. context_sample_matrix.031918.txt and SignatureAnalyzer_DBS_signature_ probability.context_sample_matrix.042018.txt). Somatic mutations observed in the PCAWG catalog were divided into three categories: SBS and DBS. Next, signature contribution values were assigned to mutations according to their category, sample identificatory and mutation feature type. For SBS signatures, the feature type consists of substitution and 1 of 96-trinucleotide contexts of the mutation, which was obtained using a MutationalPatterns R package (Blokzijl et al, 2018), whereas a DBS signature feature was a dinucleotide substitution. The mutational signature analysis was performed for each of the cancer type.

We compared the contribution of mutational signatures between regulatory mutations and the remaining somatic mutations. The direction of the signature contribution change was defined as the difference of the average signature contribution of regulatory mutations and the remaining somatic mutations. To evaluate the significance of the signature contribution differences for each of the cancer type, we performed a Kolmogorov–Smirnov two-sided test.

To measure the similarity between mutational profiles of regulatory mutations and COSMIC signatures (mutational signatures version 3 released in May 2019), we used a cosine similarity score, which was estimated using a *cos_sim_matrix()* function from the MutationalPatterns R package. The mutational profiles of regulatory mutations were a 96-trinucleotide mutation count matrix, where each column represented a cancer type and each row a trinucleotide context. To determinate the similarity level of signature profiles of regulatory mutations and COSMIC signatures, we used a threshold of 0.7 cosine similarity, which was also used by Liu et al (2019). The cosine similarity analysis was reproduced for regulatory mutations at CTCF and CEBPB motifs.

Finally, the mutational profiles of regulatory mutations at position 9 of CTCF and positions 5 and 6 of CEBPB were visualized using the *plot_96_profile()* function from the MutationalPatterns R package.

### Identification of TFs with a significant number of mutated motifs

To obtain the list of TFs that have a significant number of mutated motifs in cancer, we compared the observed number of mutated motifs of each TF to the number of mutated motifs of the same TF in the simulated sets. This test was aimed to take into account the number of motifs belonging to each TF, because TFs with a very large number of motifs are expected to acquire more mutations but

that may not result in a significant effect on the TF's function in the genome. The number of regulatory mutations in motifs of each TF was counted both in the observed set as well as in the 103 simulated sets. Notably, for the observed mutation set, only motifs enriched for regulatory mutations were considered, whereas for the simulated set, similarly to the observed set, mutations were required a change in the motif matching score no <0.3 and presence of a matching binding TF peak or DNaseI signal. In addition, in the simulated sets we discarded mutations with a significantly low functionality score (one standard deviation away from the mean of scores of the regulatory mutations annotated to the same TFs in the observed set). An empirical $P$-value was computed by comparing the frequency of regulatory mutation per TF in the observed set and the simulated sets.

### Identification of significantly mutated TF motif positions

The number of mutations at the positions of each TF motif was counted by aligning all significantly mutated motifs of the same TF across the genome. An empirical $P$-value was computed by comparing the frequency of regulatory mutations at each position of the TF motif with the frequency of mutations at the same TF motif position in the simulated sets. Only mutations with the functional evidence from the simulated sets were considered, the functional impact was evaluated as described above for significantly mutated TF motifs.

Next, we validated an association of accumulated mutations within CEBPB motifs and presence of CG dinucleotides by comparing a mutational rate of methylated CpGs within CEBPB motifs and methylated CpGs next to CEBPB motifs. The Illumina HumanMethylation450 BeadChip data were obtained from ICGA Data Portal (accessed on 13 Aug 2020) for 778 patients from 21 cancer types. We kept CpGs that were methylated ($\beta$ > 0.2) in at least one patient. Based on the methylation data, we identified CpGs within CEBPB motifs and CpGs that were the closest to CEBPB motifs. Next, we checked the mutation status of CpGs by overlapping a CpG position with C>T mutations observed in the one of the cancer types that has available methylation data. Also, we required that CpG would be methylated and mutated in the same patient. In addition, we required that mutation in CEBPB-related CpGs would be annotated to CEBPB motifs. We used Fisher's Exact test to check if the mutation rate significantly differed between CpGs inside and outside of CEBPB motif. We examined the methylation rate at position 5 of CEBPB motifs in five cell lines (Table S11). For each cell line, the active motifs were defined by overlapping CEBPB motifs and ChIP-seq peaks. The inactive motifs were defined as CEBPB motifs lacking ChIP-seq peaks from the corresponding cell line. The methylation levels for position 5 of the motifs were extracted from the ENCODE WGBS experiments. A Mann–Whitney test was performed by comparing the methylation levels of active and inactive motifs genome-wide for each cell line.

Finally, we evaluated the difference in the gene expression between mutated and non-mutated CEBPB motifs. We identified all genes located ±2 kb around mutated CEBPB motifs and divided RNA-seq samples based on mutation status of the motif. For each mutated sample, we randomly selected a sample that was not mutated on the same position of CEBPB motif and then used to test the difference in the expression of the closest gene using a Wilcoxon signed-rank test.

### Element gene assignment

Significant regulatory mutated elements across the cohorts excluding lymphoma and melanoma cohorts were merged to obtain a final list of Pan-Cancer candidates. Only elements that had at least one regulatory mutation and mutations from three different tumor samples were considered for further analysis.

We hypothesized that multiple elements located in a neighborhood region may target the same genes. As it has been experimentally shown, two different sites in the TERT promoter lead to aberrant regulation of TERT. Known protein-coding genes and lincRNAs were obtained from GENCODE v19 (Harrow et al, 2012). Each mutated element was extended by 2 kb. Extended elements that overlapped a gene were assigned to that gene. Otherwise, the first closest downstream and upstream genes to the element were assigned. Next, each element was assigned to a feature type depending on its location. Elements that overlapped noncoding regions of a gene were assigned as intronic, elements that were located 2 kb upstream of the gene were assigned as promoters and the remaining elements were assigned as intergenic. In cases of overlaps, the feature type with the largest number of mutations was assigned. Finally, the list of enriched genes was obtained by aggregating the gene-element assignments. Moreover, to avoid the impact of coding mutations on the analysis, we excluded mutated elements that overlapped gene coding sequences.

### Gene expression analysis

Gene expression counts were processed as generated by the PCAWG-3 group. Briefly, for each sample reads were aligned with TopHat2 and STAR aligners. Read counts to genes were calculated using htseq-count against the GENCODE v19. Counts were normalized using fragments per kilobase of transcript per million mapped reads (FPKM) normalization and upper quartile normalization. The final expression values were given as an average of the TopHat2 and STAR-based alignments. The expression level of each gene was compared between the mutated and non-mutated samples using a $t$ test statistic controlling for the effect of copy number variants (see below). A randomization process, similar to the one implemented in Feigin et al (2017), was applied to evaluate the significance of each $t$-value. Briefly, the mutated and not-mutated samples were combined and randomly permuted between the two classes for 100,000 iterations. A $t$-value was obtained on each iteration by comparing samples from the two randomly assigned classes. The observed $t$-value was compared to the simulated distribution of $t$-values from the permutations and empirical $P$-values were calculated. Importantly, to incorporate more expression data we took into consideration regulatory elements with one or more mutated sample. However, only genes that had mutations in at least three samples with expression data across their associated elements were considered for this analysis.

### Copy number variants impact on the gene expression

The CNVs per sample, as well as, amplification and deletion peaks from GISTIC were provided by the PCAWG-11 group (Synapse: syn8042988 and syn8293244: amp_genes.conf_95.rmcnv.pt_170207.txt

and del_genes.conf_95.rmcnv.pt_170207.txt). Only CNVs with three stars (the highest confident) were kept and intersected with the significant amplification and deletion peaks. Samples with mutations within the extracted CNV segments in the same samples were identified and removed from the set of mutated elements. The gene expression analysis on the new set of mutated elements was performed as described above.

### Pathway enrichment analysis

The KEGG pathways of *Homo sapiens* were downloaded via the REST-style KEGGAPI (Release 74.0) (Kanehisa & Goto, 2000; Kanehisa et al, 2014). The genes associated to the Pan-Cancer set (excluding lymphoma and melanoma cohorts) of recurrent elements were searched in each pathway and the enrichment was assessed based on a hypergeometric test; taking into account the total number of genes in the pathway, the number of the putative target genes enriched in the pathway, and the total number of genes used as in the test. *P*-values were adjusted using the Benjamini–Hochberg method.

## Data Availability

The PCAWG datasets used in the study are available at Synapse (https://www.synapse.org/): somatic mutations (syn9758012), RNA-seq data (syn5553991), mutational signatures (syn11738306), and somatic copy number (syn8042988 and syn8293244). The annotation data are publicly available through the ENCODE portal (http://www.encodeproject.org/).

## Supplementary Information

## Acknowledgements

We thank the members of the PCAWG project for providing the mutation calls and the RNA-seq datasets. This analysis would not have been possible without the resources provided by the ENCODE, RoadMap, and FANTOM projects. We thank Zeeshan Khaliq of our laboratory for his contributions to the analysis. We would also like to thank Inigo Martincorena from the Sanger Institute for providing the mutation simulation script. The mutation simulations were generated on the Swedish National Infrastructure for Computing through Uppsala Multidisciplinary Center for Advanced Computational Science. This work was funded by grants from Uppsala University (HM Umer), Institute of Computer Science, Polish Academy of Sciences and SYMFONIA project from Polish National Science Centre (J Komorowski), and the Swedish Cancer Foundation (No. 160518, 170296 and 180765) (C Wadelius).

### Author Contributions

HM Umer: conceptualization, data curation, software, formal analysis, validation, visualization, methodology, performed the analysis, implemented the study and wrote the manuscript, and writing—original draft, review, and editing.

K Smolinska: software, formal analysis, visualization, methodology, performed the analysis and wrote the manuscript, and writing—review and editing.

J Komorowski: conceptualization, supervision, funding acquisition, project administration, contributed in writing the manuscript and supervised the study, and writing—original draft, review, and editing.

C Wadelius: conceptualization, supervision, funding acquisition, investigation, project administration, conceived the strategy to investigate annotated cell-specific motifs, contributed in writing the manuscript and supervised the study, and writing—original draft, review, and editing.

### Conflict of Interest Statement

The authors declare that they have no conflict of interest.

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
