## [Reviewer comments · Life Science Alliance]

Life Science Alliance

Functional annotation of noncoding mutations in cancer

Husen Umer, Karolina Smolinska, Jan Komorowski, and Claes Wadelius

DOI: <https://doi.org/10.26508/lsa.201900523>

Corresponding author(s): Claes Wadelius, Science for Life Laboratory, Department of Immunology, Genetics and Pathology, Uppsala University, Uppsala, Sweden

Review Timeline:

Submission Date:	2019-08-14
Editorial Decision:	2019-10-14
Revision Received:	2020-10-15
Editorial Decision:	2020-11-20
Revision Received:	2021-02-05
Editorial Decision:	2021-06-01
Revision Received:	2021-06-18
Editorial Decision:	2021-06-21
Revision Received:	2021-06-29
Accepted:	2021-06-29

Transaction Report:

October 14, 2019

Re: Life Science Alliance manuscript #LSA-2019-00523-T

Claes Wadelius
Science for Life Laboratory, Department of Immunology, Genetics and Pathology, Uppsala University, Uppsala, Sweden

Dear Dr. Wadelius,

Thank you for submitting your manuscript entitled "Functional annotation of noncoding mutations in cancer" to Life Science Alliance. The manuscript was assessed by expert reviewers, whose comments are appended to this letter. Please excuse the delay in getting back to you, it took longer than anticipated to obtain three reports on your work.

As you will see, while the reviewers appreciate the aim of your study, they also think that the value to the community is currently compromised, especially by the lack of ensuring a low false positive rate. All three reviewers provide constructive input, however, a major revision and re-analysis is required to allow publication, the outcome of which is rather unclear at this stage. Should you be able to perform the requested re-analyses, including better control for false positives and a proper description of the methods used, we would be happy to consider a revised version for publication here. Please note, however, that we'd need strong support from the reviewers on such a revised version. So please consider your options carefully. Please let us know in case you would prefer to withdraw your manuscript at this stage.

Thank you for this interesting contribution to Life Science Alliance. We are looking forward to

receiving your revised manuscript.

Sincerely,

B. MANUSCRIPT ORGANIZATION AND FORMATTING:

Reviewer #1 (Comments to the Authors (Required)):

Umer et al search for recurrent noncoding mutations in regulatory elements.

The manuscript is overall well written, and the methods were easy to follow.

The analysis leads to the re-discovery of the well-known CTCF mutations and TERT promoter mutations and novel recurrent loci including CEBPB motifs. This is an interesting finding, but it needs more thorough analyses to strengthen the notion that the hits are functional and biological important noncoding mutation sites.

- More systematic filtering of potential false positives should be performed including potential local alignment issues and local mutational processes such as APOBEC mutagenesis (C>T and C>G mutations in TCA/T context).

- An FDR cutoff is applied to control for false-positives, but there is no demonstration of the inflation factor when performing statistical tests across thousands of sites. A qqplot and inflation factor should be included to prove that the statistical tests and significant findings are robust and not inflated

- From figure S7, it appears that the mutated CEBPB motifs are in quiescent chromatin. Are the nearby genes associated with a change in expression in those samples with the mutations?

Reviewer #2 (Comments to the Authors (Required)):

Umer and colleagues present a computational analysis of non-coding mutations in thousands of whole cancer genomes of the ICGC PCAWG project. Understanding the function of the non-coding genome and its variation is obviously an important area of research as it may lead to novel driver mutations and mechanisms of disease. This manuscript is a significant effort to decipher non-coding mutations by focusing on those that affect small DNA sequence motifs bound by transcription factors. The effort is supported by a large such whole-genome sequencing dataset (~2500) while earlier studies have mostly focused on fewer samples (hundreds). The authors should better refer to other efforts in the PCAWG project and put the novelty of their specific project into the consortium context (of which they are part of). I have identified some methodological concerns that are problematic and need to be addressed comprehensively as these affect the results and conclusions reported. Also, the manuscript would benefit from highlighting of a specific centrepiece finding and better distinguishing of some confounded cancer types (melanoma, lymphoma) from the other cancer types they analyse.

1. the authors identify 85 million TF binding motifs that likely represents a very large fraction of the genome (how much?). This poses a remarkable multiple-testing challenge. How is that addressed? There appears to be no FDR correction. On Page 19 they say that " $P < 0.05$ and $Z\text{-score} > 1$ " was used to select motifs. The lenient and non-corrected p-value cutoff leads to many false positive motifs.

2. The model to detect highly-recurrent mutated functional regulatory elements (MFREs) appears ad hoc and inflated. Why did the authors not use an existing driver discovery tool to find recurrent mutations? Many such tools are now available, including in the PCAWG project that the coauthors are part of. The consortium analyzed the same data and reported many fewer results (Rheinbay 2017, BioRxiv). The authors should show a QQ-plot to demonstrate that their discovery method is statistically balanced. Why did they use the 200bps window to merge motifs and would another window size be more robust?

3. I have a concern with ad-hoc excluding of high-impact protein-coding mutations. First, excluding these mutations leads to deflated background mutation rate and causes inflation of their recurrence model that evaluates frequency of mutations compared to a 50kbps window (i.e., the

window with mutations excluded). for this reason any regulatory regions near genes will appear more frequently mutated than they should (because the background window appears more sparsely mutated due to filters). Second, the authors use a filter on VEP impact annotations labelled as "high impact" to discard mutations, while not all protein-coding cancer driver mutations are high-impact, for example those in protein disordered regions. Then some of their frequent mutations in supposed TFBS motifs may still occur in protein-coding regions but their mutation impact is likely protein-coding rather than gene regulatory. A better strategy to address both these points would be to exclude all TF motifs overlapping coding sequence, while keeping all mutations in the dataset for more accurate estimation of background mutation rates.

4. The observation of CEBPB motif as frequently-mutated may be not as surprising and functional as the authors propose. The most common COSMIC mutational signatures (signature 1 and 5) are associated to a molecular clock of tumor age and accumulate over time mostly in the context of C>T. Mutational signature predictions are available in PCAWG and can be used directly by the authors. Are all or most CEBPB mutation assigned to aging signatures? It would be worth expanding this section of the manuscript in the context of mutational signatures. One way to test this statistically: is the set of the closest non-CEBPB bound CpGs adjacent to CEBPB motifs equally or much less mutated in real mutation data, compared to CpGs in CEBPB motifs? As a side note, the current title " Mutational signatures in TF binding sites" on P6 is misleading as it actually does not involve any analysis or discussion around mutational signatures in the sense of Alexandrov et al (pubmed/23945592).

5. it appears that the majority of signal originates from two tumor types, melanoma and lymphoma, for which known confounding processes in gene-regulatory regions are known (pubmed/27075101 and 16868548). For example, the best-powered pan-cancer cohort they call ATELM (~2000 samples, excluding melanoma and lymphoma) they only find 4,210 of 121,029 CFRMs, contrary to expectation because elements in individual cancer types would also show in the ATELM set. I would recommend clearly separating lymphoma and melanoma from the rest of the analysis and reporting findings distinctly, because of the large bias of lymphoma and melanoma mutation rates. The non-coding regions affected by mutations and their mRNA correlations in lymphoma have been described (pubmed/25903198). Aside TERT promoter that is well annotated and should not be reported here again, what is their most meaningful non-coding mutation found in other cancer type than lymphoma or melanoma? Perhaps that would make a nice example to highlight in the paper.

6. the manuscript would benefit from copy editing and more concise description of work done. I had a hard time finding the number of regions tested vs reported as significant throughout the text. The first section mentions several acronyms that are not defined (TF-motifs, TFBS, DHS, VEP). Scientific notation of p-values is flawed. They define multiple custom acronyms that make it difficult to follow (mFREs, CFRMs, ATELM). This is all small stuff but would help a reader who is less familiar with the field.

Reviewer #3 (Comments to the Authors (Required)):

Manuscript summary:

Husen et al. present their characterization of somatic non-coding variants overlaying potential transcription factor (TF) binding sites in the ICGC/TCGA Pan-Cancer Analysis of Whole Genomes (PCAWG) cohort of > 2,500 patients.

Discovery of driver mutations in cancer has generally focused on protein coding genes. In contrast, the non-coding genome poses a considerably harder problem, not only owing to its size, but also its lower sequence complexity and lack of functional annotation. Apart from the PCAWG consortium pre-print on the topic (<https://doi.org/10.1101/237313>), this study represents one of the first to try and interpret the role of non-coding regulatory variants in whole tumor genomes at this pan-cancer scale.

Combining simulated datasets with their approach for functional characterization of TF motifs (funMotif, currently also in pre-print at <https://doi.org/10.1101/683722>), the authors identify 25,271 candidate functional regulatory mutations (CFRMs). They describe enrichment of specific mutations at C/EBP and CTCF binding motifs and further cluster the identified motifs into mutated functional regulatory elements (mFREs), 399 of which appear to be recurrent (≥ 10 samples). Lastly, the authors annotate the elements with their closest gene and perform gene set enrichment analyses.

Major issues:

This reviewer appreciates the approach the authors take and believes there is plenty more to find in the non-coding genome. There is no low-hanging fruit however, as made clear in the corresponding PCAWG pre-print extensive measures need to be put in place to avoid hit lists swamped with false positives. It is exactly this point on which the current manuscript fails to convince.

This reviewer would, at least in part, have been more at ease if the manuscript reporting the funMotifs framework (currently a pre-print on BiorXiv), which is essential for CFRM annotation and characterization, had been described in a peer-reviewed publication prior to submission of this manuscript. However, it seems the basic methodology of this framework has been applied to gastrointestinal cancers and published by the authors in 2016 (<https://doi.org/10.1002/humu.23014>) and so is likely to be mature.

The current approach, this reviewer believes, is also briefly described in the PCAWG (non)-coding driver mutation pre-print (referred to as regDriver). While authors recycle the mutation simulation framework and datasets described in that preprint, they do not contrast their findings to those described in it. Crucially, a critical interpretation of their own findings is lacking, especially in light of residual inaccuracies in background models, sequencing and mapping artifacts or as-yet unmodeled local increases in the mutation burden highlighted in that pre-print.

Indeed, the last point is likely driving identification of most, if not all, recurrent elements picked up in lymphomas (IGH, MYC, BCL2, BCL7A, DNMT1, ...). All of these enhancer elements/promoters are frequently subject to off-target somatic hypermutation (with or without proximal structural variants, <https://doi.org/10.1016/j.cell.2014.11.013>). The highly focal nature of these events is not properly accounted for in the models, resulting in a high number of false positives. E.g. MYC or BCL2 overexpression in these cases is likely due to the associated balanced translocations to the IGH locus (which are not captured by filtering out copy number amplified samples).

Likewise, the identification of recurrent hits in the promoter region of RPL13A as reported here, has been ascribed to high UV-induced mutability, while TBC1D12 and WDR74 are reported to be subject to artefacts in the PCAWG consortium pre-print.

Further inadequacies in the background model likely underlie the enrichment of C/EBP (and perhaps even CTCF) motifs. The enrichment for mutation at C/EBP recognition motifs could be explained by the increased mutation rate at methylated compared to unmethylated CpG sites (Cosmic signature SBS1), together with a slight preference for binding of C/EBP type proteins to the methylated motif (<https://doi.org/10.1093/nar/gky1264>). Consequently, while authors state: "recurrent mutations at specific positions of genome-wide aligned TF-motifs may be a sign of positive selection" the more likely explanation lies in mutation rates and processes (SBS1). In addition, the reasoning of the authors for a functional impact of position 5/6 mutation in the C/EBP motif is weak. A quick look at the HOCOMOCO motif collection for C/EBPs and the realization that these are symmetric motifs would suggest that in many cases, CEBPB is likely still able to recognize its target mutated at position 5 or 6 (but not both).

A similar, yet unknown bias may explain the increased SBS17a-type (C[T>C]N, possibly oxidative damage) mutations at CTCF binding sites in gastrointestinal tumors. In addition, this enrichment at position 9 of the motif has been reported before by the authors (<https://doi.org/10.1002/humu.23014>).

Combining multiple proximal candidate mutated motifs into mutated functional regulatory elements (mFREs) is an excellent approach, and this reviewer believes this may yield good results, provided the input CRFMs are rid of the false positives highlighted above. Strong controls against local hypermutation and other biases should be put in place.

At least in part due to the underestimation of the effects of local hypermutation and non-homogeneous mutation rate, the authors pick up significantly mutated regulatory elements which are only mutated in 2 samples, suggesting inflated statistical power. Faced with 8,716 "recurrent mFREs", authors apply an ad hoc threshold of mFREs mutated in {greater than or equal to} 10 samples without argumentation, instead of amending the model

Lastly, as a considerable number of false positives contribute to the input, gene set enrichment is unlikely to yield meaningful results. The significant enrichment of "KEGG Cancer Pathways" annotations will to a large extent be influenced by false positive, recurrent hits like IGHJ6, BCL2, MYC, BCL7A, DNMT1, BTG2 ... from the lymphoma cohort (cfr Table S4)

Additional questions, remarks and issues:

- Have the authors excluded the (few) multisample cases present in the PCAWG cohort, as these will share mutations, potentially biasing results. Only the exclusion of non-whitelisted (and hypermutated) samples is mentioned.
- It is not clear why lymphoma and melanoma samples are excluded from the pan-cancer analysis. The authors mention this allows them to account for the high mutation rates in these tumor types. However, when looking at per-sample mutation burden (Figure S1 and Table S1), esophageal, lung and bladder cancer samples exhibit similarly high mutational loads. In addition, 63 samples with >100,000 mutations have already been filtered out.
- Supplementary Figures 1,2,3,5 and 6 would be more readily interpretable with a log₁₀-scaled y-axis.
- VEP should be spelled out in full (and maybe briefly elaborated on) upon first use.
- Authors identify CFRMs in part through a significant effect of the mutation on the motif score. While in most cases one would expect this to be a negative effect, it can in rare cases also be positive. Do the authors find any such gain-of-function-like cases?

- The legend of Fig 1. Is a bit wordy
- Accumulative change to cumulative
- Clarify that the GM12878 is a lymphoblastoid cell line upon first mention.
- "downreagulated" should be corrected to downregulated
- These probably reflect a single intended sentence: "Overall, we processed more than 25 million mutations identified in the genomes of 2,515 cancer-matching normal samples across 27 cancer types. utilized 2.6 billion simulated mutations and millions of functional annotation tracks were utilized."
- DNase1 should be corrected to DNaseI
- In the methods section on "Motif Identification and Annotation", it is unclear where the Z-scores are coming from.
- A threshold of 0.3 is applied as the minimal difference in matching score between wild-type and mutated motif. Is there any statistical underpinning for this threshold?
- Additional details would be helpful on the statistical testing performed to identify TFs with a significant number of mutated motifs
- "execrated" should be corrected to extracted
- A t-test is performed comparing the methylation levels of active and inactive motifs genome-wide for each cell line. While this should be robust, a non-parametric test might be better suited.
- In the pathway enrichment section, the phrase "number of genes in the universe" may be rephrased.
- While filtering samples based on copy number to assess the impact on gene expression, the authors only keep segments with 3 stars. It is worth noting that the precise copy number of amplifications is generally difficult to fit and few of these segments will have been assigned a 3-star rating. As a result, the authors are the opposite of conservative in filtering out samples with amplifications.

Reviewer #1:

Umer et al search for recurrent noncoding mutations in regulatory elements. The manuscript is overall well written, and the methods were easy to follow. The analysis leads to the re-discovery of the well-known CTCF mutations and TERT promoter mutations and novel recurrent loci including CEBPB motifs. This is an interesting finding, but it needs more thorough analyses to strengthen the notion that the hits are functional and biological important noncoding mutation sites.

- More systematic filtering of potential false positives should be performed including potential local alignment issues and local mutational processes such as APOBEC mutagenesis (C>T and C>G mutations in TCA/T context).

We have added more rigorous testing to increase confidence in the findings and remove false positive identifications.

- All samples that were hyper-mutated (>100k mutations) were excluded from the analysis.
- The regulatory score of each mutations is now compared to scores of simulated mutations in the same genomic region (50kb window) and the calculated p-values are adjusted for multiple testing.
- Moreover, mutated regulatory elements that have a significant score are further evaluated to test for mutational burden using ActiveDriverWGS and only those with FDR<0.05 are reported. These modifications reduced the number of highly recurrent elements from 399 to 77.

- An FDR cutoff is applied to control for false-positives, but there is no demonstration of the inflation factor when performing statistical tests across thousands of sites. A qqplot and inflation factor should be included to prove that the statistical tests and significant findings are robust and not inflated

We have conducted tests to check the validity of our statistical approach. The QQ-plots shown in new Supplementary figures 4 and 5 demonstrate that the p-values follow the p-values from the null distribution.

- From figure S7, it appears that the mutated CEBPB motifs are in quiescent chromatin. Are the nearby genes associated with a change in expression in those samples with the mutations?

We have performed the test by comparing nearby genes in mutated and non-mutated samples. Genes near mutated CEBPB motifs have a significant difference in expression (Wilcoxon signed-rank test, P-value<0.05).

Reviewer #2:

Umer and colleagues present a computational analysis of non-coding mutations in thousands of whole cancer genomes of the ICGC PCAWG project. Understanding the function of the non-coding genome and its variation is obviously an important area of research as it may lead to novel driver mutations and mechanisms of disease. This manuscript is a significant effort to decipher non-coding mutations by focusing on those that affect small DNA sequence motifs bound by transcription factors. The effort is supported by a large such whole-genome sequencing dataset (~2500) while earlier studies have mostly focused on fewer samples (hundreds).

The authors should better refer to other efforts in the PCAWG project and put the novelty of their specific project into the consortium context (of which they are part of). I have identified some methodological concerns that are problematic and need to be addressed comprehensively as these affect the results and conclusions reported. Also, the manuscript would benefit from highlighting of a specific centrepiece finding and better distinguishing of some confounded cancer types (melanoma, lymphoma) from the other cancer types they analyse.

1. the authors identify 85 million TF binding motifs that likely represents a very large fraction of the genome (how much?). This poses a remarkable multiple-testing challenge. How is that addressed? There appears to be no FDR correction. On Page 19 they say that " $P < 0.05$ and $Z\text{-score} > 1$ " was used to select motifs. The lenient and non-corrected p-value cutoff leads to many false positive motifs.

We thank the reviewer for this helpful insight, we absolutely agree that proper p-value correction was necessary to ensure integrity of the significant motifs. Thus, we have modified our workflow and corrected all p-values considering the large set of motifs tested (covering 14% of the genome) and $FDR < 0.05$ is used to select motifs with a significant functionality score (See Methods)

2. The model to detect highly-recurrent mutated functional regulatory elements (MFREs) appears ad hoc and inflated. Why did the authors not use an existing driver discovery tool to find recurrent mutations? Many such tools are now available, including in the PCAWG project that the coauthors are part of. The consortium analyzed the same data and reported many fewer results (Rheinbay 2017, BioRxiv). The authors should show a QQ-plot to demonstrate that their discovery method is statistically balanced. Why did they use the 200bps window to merge motifs and would another window size be more robust?

We have employed ActiveDriverWGS from the PCAWG project to test mutational burden of the identified regulatory elements and only those with $FDR < 0.05$ were retained. We would like to note that only elements that had a significant functionality score from relevant cancer types were selected for the test. We have added a QQ-plot demonstrating the p-values of the identified elements from ActiveDriverWGS follow P-values from the null (normal) distribution (Supplementary Figure 5).

We used 200bps to form regulatory elements based on our previous findings (Diamanti et al 2016 NAR) where we showed more than 86% of nearby TFBSs have their peak summits within 200bps of each other therefore it seemed to be a robust threshold to define

regulatory elements. Also data from DNaseI hypersensitive sites in the latest release from the ENCODE project show that the average peak width i.e. average size of a regulatory element is 203 bp (Vierstra et al Nature 583:729).

3. I have a concern with ad-hoc excluding of high-impact protein-coding mutations. First, excluding these mutations leads to deflated background mutation rate and causes inflation of their recurrence model that evaluates frequency of mutations compared to a 50kbp window (i.e., the window with mutations excluded). For this reason any regulatory regions near genes will appear more frequently mutated than they should (because the background window appears more sparsely mutated due to filters). Second, the authors use a filter on VEP impact annotations labelled as "high impact" to discard mutations, while not all protein-coding cancer driver mutations are high-impact, for example those in protein disordered regions. Then some of their frequent mutations in supposed TFBS motifs may still occur in protein-coding regions but their mutation impact is likely protein-coding rather than gene regulatory. A better strategy to address both these points would be to exclude all TF motifs overlapping coding sequence, while keeping all mutations in the dataset for more accurate estimation of background mutation rates.

Thanks for the suggestions, we have followed the outlined strategy i.e. all mutations are kept during in all statistical tests to keep the correct mutation rates. Afterwards, prior to gene and pathway enrichment analysis we remove all mutated motifs that overlap protein coding genes.

4. The observation of CEBPB motif as frequently-mutated may be not as surprising and functional as the authors propose. The most common COSMIC mutational signatures (signature 1 and 5) are associated to a molecular clock of tumor age and accumulate over time mostly in the context of C>T. Mutational signature predictions are available in PCAWG and can be used directly by the authors. Are all or most CEBPB mutations assigned to aging signatures? It would be worth expanding this section of the manuscript in the context of mutational signatures. One way to test this statistically: is the set of the closest non-CEBPB bound CpGs adjacent to CEBPB motifs equally or much less mutated in real mutation data, compared to CpGs in CEBPB motifs? As a side note, the current title "Mutational signatures in TF binding sites" on P6 is misleading as it actually does not involve any analysis or discussion around mutational signatures in the sense of Alexandrov et al (pubmed/23945592).

We have added a new section showing the mutational signatures of the regulatory mutations and their differential contributions in comparison to non-regulatory mutations. We confirm that around 90% of the CEBPB mutations are assigned to age signature (SBS1) since the mutations are mostly at position 5 and 6 of the motif which is a CpG site. Analysis of methylation data from the same tumors indicated a significantly lower mutation rate at methylated CG dinucleotides within CEBPB motifs in comparison to neighboring methylated CpG sites (Fisher's Exact test, P-value= 0.012, Odds ratio=3.83). Also, we observed that the active motifs of CEBPB are significantly less methylated genome-wide than inactive motifs and still there is an abundance of mutations at position 5 (Mann-Whitney test, P-value = 3.33e-239).

Moreover, a recent preprint by Ershova et al (bioRxiv 2020, doi: 10.1101/2020.06.11.146175) shows enhanced binding of CEBPs to C>T mutations. They also suggest that the enhanced binding facilitates fixation of the mutations by limiting accessibility of the motif for DNA repair.

We would like to clarify here that the aim of our study is to identify mutations that have an impact on TF binding in cancer. Thus, our findings indicate the possible roles of the regulatory mutations despite the underlying mechanisms that may explain the source of the mutations.

5. it appears that the majority of signal originates from two tumor types, melanoma and lymphoma, for which known confounding processes in gene-regulatory regions are known (pubmed/27075101 and 16868548). For example, the best-powered pan-cancer cohort they call ATELM (~2000 samples, excluding melanoma and lymphoma) they only find 4,210 of 121,029 CFRMs, contrary to expectation because elements in individual cancer types would also show in the ATELM set. I would recommend clearly separating lymphoma and melanoma from the rest of the analysis and reporting findings distinctly, because of the large bias of lymphoma and melanoma mutation rates. The non-coding regions affected by mutations and their mRNA correlations in lymphoma have been described (pubmed/25903198). Aside TERT promoter that is well annotated and should not be reported here again, what is their most meaningful non-coding mutation found in other cancer type than lymphoma or melanoma? Perhaps that would make a nice example to highlight in the paper.

Thank you for suggestions, we have now completely separated/removed results from melanoma and lymphoma cohorts.

The new findings reported here are the high enrichment rate of mutations in CEBPB motifs as well as the significant enrichment of cancer-associated pathways for regulatory mutations.

6. the manuscript would benefit from copy editing and more concise description of work done. I had a hard time finding the number of regions tested vs reported as significant throughout the text. The first section mentions several acronyms that are not defined (TF-motifs, TFBS, DHS, VEP). Scientific notation of p-values is flawed. They define multiple custom acronyms that make it difficult to follow (mFREs, CFRMs, ATELM). This is all small stuff but would help a reader who is less familiar with the field.

We have thoroughly revised the text and fixed the issues raised by the reviewer.

We have added details regarding the number of mutations and elements per cohort in Supplementary Table 7.

Reviewer #3:

Manuscript summary:

Husen et al. present their characterization of somatic non-coding variants overlaying potential transcription factor (TF) binding sites in the ICGC/TCGA Pan-Cancer Analysis of Whole Genomes (PCAWG) cohort of > 2,500 patients.

Discovery of driver mutations in cancer has generally focused on protein coding genes. In contrast, the non-coding genome poses a considerably harder problem, not only owing to its size, but also its lower sequence complexity and lack of functional annotation. Apart from the PCAWG consortium pre-print on the topic (<https://doi.org/10.1101/237313>), this study represents one of the first to try and interpret the role of non-coding regulatory variants in whole tumor genomes at this pan-cancer scale.

Combining simulated datasets with their approach for functional characterization of TF motifs (funMotif, currently also in pre-print at <https://doi.org/10.1101/683722>), the authors identify 25,271 candidate functional regulatory mutations (CFRMs). They describe enrichment of specific mutations at C/EBP and CTCF binding motifs and further cluster the identified motifs into mutated functional regulatory elements (mFREs), 399 of which appear to be recurrent (≥ 10 samples). Lastly, the authors annotate the elements with their closest gene and perform gene set enrichment analyses.

Major issues:

This reviewer appreciates the approach the authors take and believes there is plenty more to find in the non-coding genome. There is no low-hanging fruit however, as made clear in the corresponding PCAWG pre-print extensive measures need to be put in place to avoid hit lists swamped with false positives. It is exactly this point on which the current manuscript fails to convince.

This reviewer would, at least in part, have been more at ease if the manuscript reporting the funMotifs framework (currently a pre-print on BiorXiv), which is essential for CFRM annotation and characterization, had been described in a peer-reviewed publication prior to submission of this manuscript. However, it seems the basic methodology of this framework has been applied to gastrointestinal cancers and published by the authors in 2016 (<https://doi.org/10.1002/humu.23014>) and so is likely to be mature.

Indeed the current approach is based on the initial method that was applied in our analysis of gastrointestinal cancers. Here, we have further developed to include more tissue types and tumor types to perform a more comprehensive analysis of noncoding mutations in transcription factor binding sites.

The current approach, this reviewer believes, is also briefly described in the PCAWG (non)-coding driver mutation pre-print (referred to as regDriver). While authors recycle the mutation simulation framework and datasets described in that preprint, they do not contrast their findings to those described in it. Crucially, a critical interpretation of their own

findings is lacking, especially in light of residual inaccuracies in background models, sequencing and mapping artifacts or as-yet unmodeled local increases in the mutation burden highlighted in that pre-print.

In the revised version we have related our work to the findings from PCAWG with showing the distinctions of the approach applied here.

Indeed, the last point is likely driving identification of most, if not all, recurrent elements picked up in lymphomas (IGH, MYC, BCL2, BCL7A, DNMT1, ...). All of these enhancer elements/promoters are frequently subject to off-target somatic hypermutation (with or without proximal structural variants, <https://doi.org/10.1016/j.cell.2014.11.013>). The highly focal nature of these events is not properly accounted for in the models, resulting in a high number of false positives. E.g. MYC or BCL2 overexpression in these cases is likely due to the associated balanced translocations to the IGH locus (which are not captured by filtering out copy number amplified samples).

We have modified our workflow to compare the regulatory score of the observed mutations to nearby simulated mutations within a 50KB window. In comparison to our previous approach that was genome-wide, this modification controls for local hyper-mutation rates. Also, the p-values that are obtained for the regulatory mutations are adjusted for multiple testing to further limit the false positive rate.

We have conducted the analysis on the melanoma and lymphoma cohorts independent of the other cohorts, also the large cohort (ATELM) doesn't contain any samples from lymphoma and melanoma in order to avoid the effects of these two cohorts on the main results.

Likewise, the identification of recurrent hits in the promoter region of RPL13A as reported here, has been ascribed to high UV-induced mutability, while TBC1D12 and WDR74 are reported to be subject to artefacts in the PCAWG consortium pre-print.

Indeed, our current set of element no longer contain elements of TBC1D12, and our results also suggest that WDR74 mutations have no regulatory roles.

Further inadequacies in the background model likely underlie the enrichment of C/EBP (and perhaps even CTCF) motifs. The enrichment for mutation at C/EBP recognition motifs could be explained by the increased mutation rate at methylated compared to unmethylated CpG sites (Cosmic signature SBS1), together with a slight preference for binding of C/EBP type proteins to the methylated motif (<https://doi.org/10.1093/nar/gky1264>). Consequently, while authors state: "recurrent mutations at specific positions of genome-wide aligned TF-motifs may be a sign of positive selection" the more likely explanation lies in mutation rates and processes (SBS1). In addition, the reasoning of the authors for a functional impact of position 5/6 mutation in the C/EBP motif is weak. A quick look at the HOCOMOCO motif collection for C/EBPs and the realization that these are symmetric motifs would suggest that in many cases, CEBPB is likely still able to recognize its target mutated at position 5 or 6 (but not both).

In agreement with recent results shared in a preprint by Ershova et al (bioRxiv 2020, doi: 10.1101/2020.06.11.146175) mutations at the CpG site seem to further enhance binding of

CEBPs. These results indicate that that mutations do have regulatory effects which in this case is enhancement of binding.

A similar, yet unknown bias may explain the increased SBS17a-type (C[T>C]N, possibly oxidative damage) mutations at CTCF binding sites in gastrointestinal tumors. In addition, this enrichment at position 9 of the motif has been reported before by the authors (<https://doi.org/10.1002/humu.23014>).

We have added a new section describing the contribution of mutational signatures in all regulatory mutations, mutations in CEBPB motifs and mutations in CTCF motifs. Indeed we do see an over-representation of SBS17b for CTCF mutations in gastrointestinal and lymph-NOS cancers (Supplementary Figure 9). Thus we agree that they could be explained by oxidative damage of DNA. Of note, this signature is not observed in the other cancer types which could indicate different underlying mechanisms.

Combining multiple proximal candidate mutated motifs into mutated functional regulatory elements (mFREs) is an excellent approach, and this reviewer believes this may yield good results, provided the input CRFMs are rid of the false positives highlighted above. Strong controls against local hypermutation and other biases should be put in place.

At least in part due to the underestimation of the effects of local hypermutation and non-homogeneous mutation rate, the authors pick up significantly mutated regulatory elements which are only mutated in 2 samples, suggesting inflated statistical power. Faced with 8,716 "recurrent mFREs", authors apply an ad hoc threshold of mFREs mutated in {greater than or equal to} 10 samples without argumentation, instead of amending the model

To overcome the issues that lead to false positives, we have also employed a mutation burden test (ActiveDriverWGS from PCAWG) to evaluate mutation significance of the identified elements. Also, our aim in focusing on elements mutated in more than 10 samples is to provide a more detailed analysis of highly recurrent elements however we want to clarify that we think even those mutated in two samples can have regulatory effects.

Lastly, as a considerable number of false positives contribute to the input, gene set enrichment is unlikely to yield meaningful results. The significant enrichment of "KEGG Cancer Pathways" annotations will to a large extent be influenced by false positive, recurrent hits like IGJH6, BCL2, MYC, BCL7A, DNMT1, BTG2 ... from the lymphoma cohort (cfr Table S4)

We would like to clarify that elements and genes from the lymphoma and melanoma cohorts are excluded in performing the pathway enrichment analysis. Thus, the enriched pathways only account for results from the other cohorts. Additionally, regulatory mutations in coding regions are also excluded to avoid inclusion of coding mutations and only include genes based on mutations in their regulatory elements.

Additional questions, remarks and issues:

- Have the authors excluded the (few) multisample cases present in the PCAWG cohort, as these will share mutations, potentially biasing results. Only the exclusion of non-whitelisted (and hypermutated) samples is mentioned.

We have used the mutation dataset generated by the driver group in PCAWG. Only a single aliquot was assigned to each sample; in cases where multiple aliquots were present, a single aliquot was selected based on the following criteria, in order of importance:

- we prioritized primary tumors over metastatic or recurrent tumors
- we selected aliquots with an OxoG score higher than 40
- we prioritized aliquots with the highest quality (as indicated by the Stars values)
- we prioritized aliquots with RNA-seq data availability
- we prioritized aliquots with the lowest contamination (as indicated by the ContEst values)
- if a selection could not be made after applying the above filters we selected an aliquot randomly

(as stated in Supplementary Methods by Rheinbay et al Nature 2020).

- It is not clear why lymphoma and melanoma samples are excluded from the pan-cancer analysis. The authors mention this allows them to account for the high mutation rates in these tumor types. However, when looking at per-sample mutation burden (Figure S1 and Table S1), esophageal, lung and bladder cancer samples exhibit similarly high mutational loads. In addition, 63 samples with >100,000 mutations have already been filtered out. Melanoma was handled separately due to its high mutation rate compared to all other cancers. However, the reason for handling lymphomas distinctly was due to the local hypermutation rates at the IGH locus. We have now clarified this in the main text.

- Supplementary Figures 1,2,3,5 and 6 would be more readily interpretable with a log₁₀-scaled y-axis.

They are now in log₁₀ scale.

- VEP should be spelled out in full (and maybe briefly elaborated on) upon first use.

It is not used anymore so it is removed from the text.

- Authors identify CFRMs in part through a significant effect of the mutation on the motif score. While in most cases one would expect this to be a negative effect, it can in rare cases also be positive. Do the authors find any such gain-of-function-like cases?

We surely see many such examples and our method does allow for that because it computes the absolute value of the effect that could increase or decrease the affinity.

- The legend of Fig 1. Is a bit wordy

We have shortened the text however to

- Accumulative change to cumulative

- Clarify that the GM12878 is a lymphoblastoid cell line upon first mention.

- "downregulated" should be corrected to downregulated

- These probably reflect a single intended sentence: "Overall, we processed more than 25 million mutations identified in the genomes of 2,515 cancer-matching normal samples across 27 cancer types. utilized 2.6 billion simulated mutations and millions of functional annotation tracks were utilized."

- DNase1 should be corrected to DNaseI

- In the methods section on "Motif Identification and Annotation", it is unclear where the Z-scores are coming from.

The suggested modifications above are implemented in the main text.

- A threshold of 0.3 is applied as the minimal difference in matching score between wild-type and mutated motif. Is there any statistical underpinning for this threshold?

The threshold of 0.3 was set since the background nucleotide frequency for a single position is 0.25 per nucleotide.

- Additional details would be helpful on the statistical testing performed to identify TFs with a significant number of mutated motifs

This is done.

- "execrated" should be corrected to extracted

This is done.

- A t-test is performed comparing the methylation levels of active and inactive motifs genome-wide for each cell line. While this should be robust, a non-parametric test might be better suited.

We replaced the t-test with a Mann-Whitney test.

- In the pathway enrichment section, the phrase "number of genes in the universe" may be rephrased.

This is done.

- While filtering samples based on copy number to assess the impact on gene expression, the authors only keep segments with 3 stars. It is worth noting that the precise copy number of amplifications is generally difficult to fit and few of these segments will have been assigned a 3-star rating. As a result, the authors are the opposite of conservative in filtering out samples with amplifications.

We used 3 stars segments in combination with amplification and deletion peaks from GISTIC to obtain the most reliable set of CNV regions and avoid false positive CNVs.

November 20, 2020

Re: Life Science Alliance manuscript #LSA-2019-00523-TR

Prof. Claes Wadelius
Science for Life Laboratory, Department of Immunology, Genetics and Pathology, Uppsala
University, Uppsala, Sweden
BMC Husargatan 3
Uppsala 75122
Sweden

Dear Dr. Wadelius,

Thank you for submitting your revised manuscript entitled "Functional annotation of noncoding mutations in cancer" to Life Science Alliance. The manuscript has been seen by the original reviewers whose comments are appended below. While the reviewers continue to be overall positive about the work in terms of its suitability for Life Science Alliance, some important issues remain.

As you will note from the appended reviews, the reviewers are concerned that the statistics are highly inflated. It is LSA's policy to only allow one round of revision, however, given the interest from 2 of the 3 reviewers and the clear roadmap provided by the reviewers to improve the manuscript, we can make an exception in this case to allow a 2nd round of revisions. You can choose to use the approach suggested by Reviewer 2 or Reviewer 1 for the revisions. Please note that we will be sending the revised manuscript back to the referees and will need their absolute strong support to move forward.

Please submit the final revision within one month, along with a letter that includes a point by point response to the remaining reviewer comments. Let me know if this timeline is not feasible and if you need an extension.

B. MANUSCRIPT ORGANIZATION AND FORMATTING:

Sincerely,

Shachi Bhatt, Ph.D.
Executive Editor
Life Science Alliance
<https://www.lsjournal.org/>
Tweet @SciBhatt @LSAJournal

Reviewer #1 (Comments to the Authors (Required)):

Umer et al has answered comment 1 and 3. However, the statistics appear quite inflated for several tumors in the qqplots SFig 4 are 5. A non-standard Pearson correlation is used here but the genomic inflation factor is the correct approach, calculated using obs and expected pvalue quantiles with the lambda function (median of obs/exp chi-square dist). A lambda for each Tumor-analysis needs to be provided - and any inflated/deflated analyses should either be redone with more stringent criteria or excluded.

Reviewer #2 (Comments to the Authors (Required)):

The authors have improved their manuscript and addressed the majority of my comments. The manuscript has been revised thoroughly and reads much better. Their findings also appear more robust. I would recommend acceptance after addressing these final revisions.

statistical comment:

As the central finding, the authors show that CEBPB motifs are more frequently mutated in cancer than expected. The main quantitative evidence, the calculation of p-values for reflecting the enrichment of mutations in specific classes of motifs, may be flawed since the reported FDR-values are very very small. In the following example, 1777 mutations are expected and 3909 are observed and the FDR is reported as zero, whereas only 8.9% of motifs had 2+ samples with mutations.

A) P7: CEBPB motif (n= 3,909 and FDR=0 in the ALTEM cohort; Fold-enrichment=2.2) (Fig. 3b, c). 8.9% of the motifs were mutated in two or more samples.

Two more examples: B) In the ATELM cohort, 7,068 CTCF motifs were mutated (FDR=4.3e-187); C) In melanomas 1,092 CTCF motifs were mutated (FDR=2.1e-228).

According to Methods, P 20: "A p value was computed by comparing the observed mutation frequency per TF with its matching mean and standard deviation in the simulated sets. The p values were corrected for the number of TFs tested."

This method seems highly inflated and is not the right way to conduct a permutation test. The current approach assumes that the mutation frequencies from their simulated data are normally distributed and furthermore, it assumes that their small number of 103 simulations fairly captures the underlying variation (standard deviation).

The empirical p-value of a permutation test is the number of times a simulated value (here, a motif mutation frequency seen in simulated variant calls) exceeds a true value (here, the true motif mutation frequency). If the authors observe none of their 103 simulated variant callsets exceeding the true mutation frequency, the reported P should be " $P < 1/103$ " and not "FDR = 0". Note that further FDR correction on these simulation-based empirical p-values would not be required. It seems reasonable that the authors run the test on a larger number of simulations to achieve more reliable permutation p-values. It is quite likely that the major conclusions would not be affected greatly, but the current astronomy-scale p-values are misleading and should be fixed.

Minor wording/introduction/discussion comments:

The much updated section on mutation signatures and tf motifs, "Mutational patterns in TFBSs", is very interesting but it is quite difficult to read. It covers novel findings (CEBPB), confirms earlier findings (CTCF) and also ties in an interesting methylation angle. As such, the text would benefit from some strategic editing, structuring as paragraphs and careful stating of main findings.

The authors may want to balance the discussion explicitly between the passive role of mutational signatures (eg. SBS1 and SBS5) generating these motif mutations genome-wide vs. the role of positive selection and functional consequences of mutations in a smaller subset of important motifs at the regulatory elements of cancer genes.

Reviewer #3 (Comments to the Authors (Required)):

Manuscript summary:

In a revised manuscript, Husen et al. have modified their workflows characterizing somatic non-coding variants overlaying potential transcription factor (TF) binding sites in whole-genome sequenced tumor samples. While the approach has, without a doubt, improved over the previous version, the statistical models are still far from well-calibrated. One needs only to inspect the QQ-plots in Figures S4 and 5 to observe that the p-values which are obtained by the model are massively inflated. As the results from these models form the basis of the downstream analyses described in the manuscript, this reviewer is hesitant to accept the findings and conclusions presented by the authors.

Indeed, despite being the target of local hypermutation, MYC and BCL2 overexpression in Lymph-BNHL are still mentioned and linked to recurrent mutations in their first introns and promoter regions. As highlighted during the previous revision, these are known translocation targets and the overexpression is generally driven by IGH promoter hijacking.

A number of smaller specific issues also remain:

- To support their findings, the authors refer to a recent finding of C>T mismatches in C/EBP binding motifs increasing its binding affinity (Ershova et al, 2020, BioRxiv). Such mismatches however are unlikely to persist during the cell cycle. The T:G (or A:C) mismatch will give rise to one daughter strand with a C:G and another with a T:A base pair, fixating the mutation and resolving the mismatch.
- The analysis of RNA-expression dysregulation is flawed as it takes copy number, but not tumor purity, into account and the test is a two-tailed test (as far as this reviewer can deduce from the methods), suggesting no directionality of the dysregulation.
- The threshold determining the final set of recurrent single elements is still arbitrarily set at 10 or more samples.
- References supporting statements such as "EGR1 has been shown to down-regulate TERT expression and it has been suggested as a tumor suppressor. Creation of a de novo ETS motif at the EGR1 motif locus has been shown to upregulate TERT expression." are lacking.

Reviewer #1:

Umer et al has answered comment 1 and 3. However, the statistics appear quite inflated for several tumors in the qqplots SFig 4 are 5. A non-standard Pearson correlation is used here but the genomic inflation factor is the correct approach, calculated using obs and expected pvalue quantiles with the lambda function (median of obs/exp chi-square dist). A lambda for each Tumor-analysis needs to be provided - and any inflated/deflated analyses should either be redone with more stringent criteria or excluded.

We highly appreciate your suggestion on using the genomic inflation factor. Indeed, the lambda values showed clear indications of inflation in many of our cohorts. However, after implementing the suggestions provided by reviewer 2, we could clearly see the effect of our new approach on the genomic inflation factor and the qq-plots. We have provided the lambda values for all cohorts in Supplementary Table 1 and qq-plots in Supplementary figure 4. Since the lambda values of all cohorts were below or around 1, and the qq-plots were not inflated therefore we decided to keep them all in the analysis.

Additionally, we have excluded lymphoma and melanoma cohorts from the pan-cancer analysis to avoid any potential bias from hyper mutated samples and elements.

Reviewer #2:

The authors have improved their manuscript and addressed the majority of my comments. The manuscript has been revised thoroughly and reads much better. Their findings also appear more robust. I would recommend acceptance after addressing these final revisions.

statistical comment:

As the central finding, the authors show that CEBPB motifs are more frequently mutated in cancer than expected. The main quantitative evidence, the calculation of p-values for reflecting the enrichment of mutations in specific classes of motifs, may be flawed since the reported FDR-values are very very small. In the following example, 1777 mutations are expected and 3909 are observed and the FDR is reported as zero, whereas only 8.9% of motifs had 2+ samples with mutations.

A) P7: CEBPB motif ($n = 3,909$ and $FDR = 0$ in the ALTEM cohort; $Fold-enrichment = 2.2$) (Fig. 3b, c). 8.9% of the motifs were mutated in two or more samples.

Two more examples: B) In the ATELM cohort, 7,068 CTCF motifs were mutated

($FDR = 4.3e-187$); C) In melanomas 1,092 CTCF motifs were mutated ($FDR = 2.1e-228$).

According to Methods, P 20: "A p value was computed by comparing the observed mutation frequency per TF with its matching mean and standard deviation in the simulated sets. The p values were corrected for the number of TFs tested."

This method seems highly inflated and is not the right way to conduct a permutation test. The current approach assumes that the mutation frequencies from their simulated data are normally distributed and furthermore, it assumes that their small number of 103 simulations fairly captures the underlying variation (standard deviation).

The empirical p-value of a permutation test is the number of times a simulated value (here, a motif mutation frequency seen in simulated variant calls) exceeds a true value (here, the true motif mutation frequency). If the authors observe none of their 103 simulated variant callsets exceeding the true mutation frequency, the reported P should be " $P < 1/103$ " and not " $FDR = 0$ ". Note that further FDR correction on these simulation-based empirical p-values would not be required. It seems reasonable that the authors run the test on a larger number of simulations to achieve more reliable permutation p-values. It is quite likely that the major conclusions would not be affected greatly, but the current astronomy-scale p-values are misleading and should be fixed.

Thank you so much for your effort in describing in detail the strategy that helped us to improve our methodology. Based on your suggestion, we calculated empirical p-values for evaluating the significance of the mutated regulatory elements as well as all transcription factor motifs since the simulations are used in both analyses.

In brief, for calculating p-values for each transcription factor motif, we counted how many times the TF motif has a mutation frequency higher than the mutation frequency in the observed set. We applied the same method to calculate p-values for motif positions.

Also, a p-value was calculated for each regulatory element based on the number of elements in the simulated sets with a higher score. The empirical p-values calculated following this approach yielded much lower inflation than our previous p-values in the qq-plots as measured by the genomic inflation factor (suggested by reviewer 1).

Minor wording/introduction/discussion comments:

The much updated section on mutation signatures and tf motifs, "Mutational patterns in TFBSs", is very interesting but it is quite difficult to read. It covers novel findings (CEBPB), confirms earlier findings (CTCF) and also ties in an interesting methylation angle. As such, the text would benefit from some strategic editing, structuring as paragraphs and careful stating of main findings.

We have updated the entire section to make it easier for the reader to follow.

The authors may want to balance the discussion explicitly between the passive role of mutational signatures (eg. SBS1 and SBS5) generating these motif mutations genome-wide vs. the role of positive selection and functional consequences of mutations in a smaller subset of important motifs at the regulatory elements of cancer genes.

We have presented the contribution of SBS1 and SBS5 signatures in highly enriched regulatory mutations at positions 5 and 6 within the CEBPB motif. Our results suggested that those tumor aged-related signatures do not significantly contribute to creation of regulatory mutations and other mutational processes may play a role in creation of mutations at this motif.

Reviewer #3:

Manuscript summary:

In a revised manuscript, Husen et al. have modified their workflows characterizing somatic non-coding variants overlaying potential transcription factor (TF) binding sites in whole-genome sequenced tumor samples. While the approach has, without a doubt, improved over the previous version, the statistical models are still far from well-calibrated. One needs only to inspect the QQ-plots in Figures S4 and 5 to observe that the p-values which are obtained by the model are massively inflated. As the results from these models form the basis of the downstream analyses described in the manuscript, this reviewer is hesitant to accept the findings and conclusions presented by the authors.

Thank you for pointing out the inflation rates in our statistics, we have particularly addressed this issue in this second-revised version of our manuscript. We have modified our pipeline to calculate empirical p-values for the mutated elements by comparing the scores of the elements to the simulated elements (as suggested by reviewer 2). We then measured the inflation rate in each cohort by computing the genomic inflation factor and found that none of the cohorts suffer from inflation. As we have shown in Supplementary Figure 4, the qq-plots have greatly improved over the previous reported ones. Additionally, we have re-applied ActiveDriverWGS and only elements with $FDR < 0.05$ are kept. Also, we have conditioned the elements to contain at least one regulatory mutation with direct impact on TF binding.

Indeed, despite being the target of local hypermutation, MYC and BCL2 overexpression in Lymph-BNHL are still mentioned and linked to recurrent mutations in their first introns and promoter regions. As highlighted during the previous revision, these are known translocation targets and the overexpression is generally driven by IGH promoter hijacking.

In this revised version, we have completely separated the lymphoma cohorts to avoid the effects of hyper mutated elements on our analysis across other cancer types. All analysis are conducted and presented on the significant elements from all cancer types excluding lymphoma and melanoma cohorts. Also, after the modifications in our statistical analysis, only 23 elements were identified in the lymphoma cohorts, and MYC and BCL2 were no longer among the significant elements.

A number of smaller specific issues also remain:

- To support their findings, the authors refer to a recent finding of C>T mismatches in C/EBP binding motifs increasing its binding affinity (Ershova et al, 2020, BioRxiv). Such mismatches however are unlikely to persist during the cell cycle. The T:G (or A:C) mismatch

will give rise to one daughter strand with a C:G and another with a T:A base pair, fixating the mutation and resolving the mismatch.

We have modified the text to correctly refer to C>T mutations not C>T mismatches.

- The analysis of RNA-expression dysregulation is flawed as it takes copy number, but not tumor purity, into account and the test is a two-tailed test (as far as this reviewer can deduce from the methods), suggesting no directionality of the dysregulation.

Due to the limited availability of samples with gene expression data, we were not able to incorporate tumor purity data in our study.

- The threshold determining the final set of recurrent single elements is still arbitrarily set at 10 or more samples.

Similar to the approach applied in Rheinbay et al Nat 2020, now we have only considered significant regulatory elements that are mutated in at least three tumor samples. In the final section, thus we have removed filters on highly recurrent elements. However, to present the results in a concise way we have mentioned the top elements and genes to avoid over-presenting the results. Also, we want to emphasize that the entire set of significant elements have the potential to be functional and affect gene regulation in cancer due to the stringent filters that we have applied in identifying the elements genome-wide.

- References supporting statements such as "EGR1 has been shown to down-regulate TERT expression and it has been suggested as a tumor suppressor. Creation of a de novo ETS motif at the EGR1 motif locus has been shown to upregulate TERT expression." are lacking.

We have revised the manuscript to add any missing references including citations to support this statement.

June 1, 2021

Re: Life Science Alliance manuscript #LSA-2019-00523-TRR

Prof. Claes Wadelius
Science for Life Laboratory, Department of Immunology, Genetics and Pathology, Uppsala
University, Uppsala, Sweden
BMC Husargatan 3
Uppsala 75122
Sweden

Dear Dr. Wadelius,

Thank you for submitting your revised manuscript entitled "Functional annotation of noncoding mutations in cancer" to Life Science Alliance. The manuscript has been seen by the original reviewers whose comments are appended below. While the reviewers continue to be overall positive about the work in terms of its suitability for Life Science Alliance (LSA), some important issues remain.

We have now thoroughly analyzed the revised version of the manuscript, the revised figure S4, and the pbp response to the reviewers' remaining concern. In conjunction with our academic expert, we agree with Reviewer 1 that the QQ plots (even in revised Figure S4) have no signal. We recommend that the claims dependent on these QQ plots be removed from the manuscript and a revised manuscript without these claims should be submitted. We would be happy to discuss this further over email or phone, if you would prefer.

Our general policy is that papers are considered through only one revision cycle; however, given that the suggested changes are relatively minor, we are open to one additional short round of revision. Please note that I will expect to make a final decision without additional reviewer input upon resubmission.

Please submit the final revision within one month, along with a letter that includes a point by point response to the remaining reviewer comments.

B. MANUSCRIPT ORGANIZATION AND FORMATTING:

Sincerely,

Shachi Bhatt, Ph.D.
Executive Editor
Life Science Alliance
<http://www.lsjournal.org>
Tweet @SciBhatt @LSAJournal

Reviewer #1 (Comments to the Authors (Required)):

The authors have implemented a permutation-based scheme to account for the background rate of mutations. Diagnostic plots of the updated analysis are represented in six qqplots (Fig S4). Unfortunately, these plots show clear signs of deflation, which can indicate problems in the underlying hypothesis to be tested. Moreover, I find the lambda values suspicious when comparing to the qqplots, which do not indicate lambda of 1 (a straight diagonal line), which can have a few points above the diagonal - ie where your 'hits' should be. Which formula was used to calculate the lambda? In essence, the plots show that there are no significant hits after correction, rather there are fewer significant elements compared to expected. Statistically, when you are doing a lot of tests, you will always find some iterations giving you a low p-value by chance.

Reviewer #2 (Comments to the Authors (Required)):

The authors have addressed my comments. I am happy to recommend publication. A proof-reading is recommended, though, and some annotations still look a little strange, for example "P value = 1.2×10^{-2} " should be written as "P value = 0.012" (Page 10 and elsewhere). For smaller P-values a format like 1.23×10^{-45} is more standard. Also, there is a very significant P-value 3.33×10^{-239} that should be checked in the light of my comments and comments of reviewer #1 (also on Page 10).

June 21, 2021

RE: Life Science Alliance Manuscript #LSA-2019-00523-TRRR

Prof. Claes Wadelius
Science for Life Laboratory, Department of Immunology, Genetics and Pathology, Uppsala University
BMC Husargatan 3
Uppsala 75122
Sweden

Dear Dr. Wadelius,

Thank you for submitting your revised manuscript entitled "Functional annotation of noncoding mutations in cancer". We would be happy to publish your paper in Life Science Alliance pending final revisions necessary to meet our formatting guidelines.

-please make sure all of the author names in our system match the author names in the manuscript. Please correct how the name of your co-author is displayed: Karolina Smolinska-Garbulowska in the system vs. Karolina Smolinska in the manuscript file

A. FINAL FILES:

B. MANUSCRIPT ORGANIZATION AND FORMATTING:

Sincerely,

June 29, 2021

RE: Life Science Alliance Manuscript #LSA-2019-00523-TRRRR

Prof. Claes Wadelius
Science for Life Laboratory, Department of Immunology, Genetics and Pathology, Uppsala
University, Uppsala, Sweden
BMC Husargatan 3
Uppsala 75122
Sweden

Dear Dr. Wadelius,

Thank you for submitting your Resource entitled "Functional annotation of noncoding mutations in cancer". It is a pleasure to let you know that your manuscript is now accepted for publication in Life Science Alliance. Congratulations on this interesting work.

DISTRIBUTION OF MATERIALS:

Again, congratulations on a very nice paper. I hope you found the review process to be constructive and are pleased with how the manuscript was handled editorially. We look forward to future exciting submissions from your lab.

Sincerely,
